# Soil Salinization and Ancient Hulled Wheat: A Study on Antioxidant Defense Mechanisms

**DOI:** 10.3390/plants14050678

**Published:** 2025-02-22

**Authors:** Ridvan Temizgul

**Affiliations:** Department of Biology, Faculty of Sciences, Erciyes University, 38039 Kayseri, Türkiye; rtemizgul@erciyes.edu.tr; Tel.: +90-352-207-6666 (ext. 33078)

**Keywords:** antioxidant, glycine betaine, hulled wheat, ROS, salt stress

## Abstract

Soil salinization, which is second only to soil erosion in terms of soil degradation, significantly hinders crop growth and development, leading to reduced yields. This study investigated the enzymatic and non-enzymatic antioxidant defense mechanisms of four ancient hulled wheat species under salt stress, with and without exogenous glycine betaine (0.5 mM). We aimed to assess the salt tolerance of these species and their potential for cultivation in saline/sodic soils. Our findings indicate that sodium and potassium chloride concentrations exceeding 100 mM induce significant stress in hulled wheat. However, combined salt stress (sodium and potassium chloride) reduced this stress by approximately 20–30%. Furthermore, exogenous glycine betaine supplementation almost completely alleviated the negative effects of salt stress, particularly in *Triticum boeoticum*. This species exhibited a remarkable ability to restore normal growth functions under these conditions. Our results suggest that ancient hulled wheat, especially *T. boeoticum*, may be a promising candidate for cultivation in sodium-saline soils. By supplementing with potassium fertilizers in addition to nitrogen, plants can effectively control salt influx into their cells and maintain intracellular K^+^/Na^+^ balance, thereby mitigating the adverse effects of salinity stress. This approach has the potential to increase crop yields and enhance food security in saline environments.

## 1. Introduction

Soil salinization, a major factor in soil degradation, severely restricts plant growth and reduces crop yields, impacting agriculture for millennia [1]. Salinity, a global issue affecting over 100 countries, results in the annual loss of over 730,000 hectares of arable land and significantly reduces crop yields (by 10–25%). Severe salinization can lead to soil desertification and complete crop failure [2]. Soil salinity, characterized by elevated levels of soluble salts such as sodium, calcium, magnesium, potassium, chloride, sulfate, bicarbonate, carbonate, and nitrate [3], is a growing concern due to factors such as excessive irrigation, fertilizer misuse, and industrial pollution [4]. This environmental stress not only compromises agricultural productivity but also disrupts ecosystem health. To address this challenge, developing salt-tolerant crops is crucial. High concentrations of sodium and chloride ions in soil impair plant water and nutrient uptake, leading to reduced growth and yield [5].

Salt stress, a major abiotic stressor, induces oxidative stress by increasing ROS production [6,7]. Reactive oxygen species (ROS) are byproducts of aerobic metabolism that can both damage cells and act as signaling molecules. To counteract ROS toxicity, plants have evolved a complex antioxidant defense system. This system comprises both enzymatic and non-enzymatic components. Enzymatic antioxidants, such as superoxide dismutase (SOD), catalase (CAT), and ascorbate peroxidase (APX), catalyze ROS detoxification, while non-enzymatic antioxidants, including ascorbate, glutathione, carotenoids, and flavonoids, directly scavenge ROS [8,9,10,11,12]. This can lead to damage to cellular components and hinder plant growth and development [13]. Plants have evolved various mechanisms to cope with salt stress, including the accumulation of osmolytes such as proline and glycine betaine (GB). These osmolytes function as osmoprotectants and antioxidants, helping plants maintain cellular integrity and alleviate oxidative damage [14]. The exogenous application of osmolytes can further enhance plant tolerance to salt stress by mitigating ROS-induced oxidative stress and improving photosynthetic efficiency [15,16]. A delicate balance between ROS production and antioxidant defense is crucial for plant health and survival. Understanding this complex interplay is essential for developing strategies to improve plant resilience to salt stress and enhance agricultural productivity [8,9].

Salt stress poses a significant threat to global wheat production and food security. To address this challenge, developing salt-tolerant wheat varieties and implementing appropriate agronomic practices is crucial. Ancestral hulled wheat varieties, with their inherent salt tolerance, offer a promising avenue for breeding programs. Türkiye, as a wheat gene center, possesses a rich genetic diversity that may harbor salt-tolerant varieties. These varieties, when cultivated in sodic/saline soils, could potentially outperform modern wheat varieties in terms of yield. Leveraging these genetic resources is essential for ensuring food security in increasingly saline agricultural lands. Wheat, a staple food globally, is vulnerable to salinity stress. Hulled wheat, the ancient ancestor of modern wheat, offers a valuable genetic resource. Modern wheat, bred for specific traits such as yield and nutritional quality, may have lost some of its natural salt tolerance. Moreover, there is growing consumer interest in ancient grains, including hulled wheat, due to their potential health benefits. These grains offer a healthier nutritional profile and may be better tolerated by individuals with gluten sensitivities.

The increasing human population and evolving lifestyles have led to significant environmental degradation. This has resulted in various abiotic stresses, such as salinity, drought, waterlogging, temperature extremes, light intensity fluctuations, radiation, ozone, and heavy metal toxicity. These stressors adversely affect crop productivity and yield, hindering efforts to meet the global food demand. To address this challenge, there is an urgent need to develop resilient crop varieties that can withstand adverse environmental conditions and maintain high yields [17].

Ancient wheats, such as einkorn, emmer, and spelt, have remained largely unchanged for centuries [18,19]. These hulled wheat species, which require milling to remove their hulls, are distinct from modern wheats [20,21]. Einkorn, a diploid wheat, is one of the earliest domesticated crops [22]. Emmer, a tetraploid wheat, is a domesticated form of wild emmer [23]. Spelt, a hexaploid wheat, is an ancestor of modern common wheat, resulting from a natural hybridization event. These ancient wheats represent a bridge between modern and wild wheat varieties. Hulled wheats, such as emmer wheat, are often overlooked and underutilized crops despite their potential for food security and cultural significance. These species, which are frequently neglected by researchers and policymakers, face the threat of extinction due to various factors. While emmer wheat constitutes less than 1% of global wheat cultivation [24], its suitability for organic farming in marginal areas has been recognized [25].

To assess the potential of ancient Turkish hulled wheat species for saline agriculture, we investigated their antioxidant responses to varying salt stress levels. We also explored the role of exogenous GB in mitigating salt stress. Enzymatic and non-enzymatic antioxidant defense mechanisms were examined in both the roots and shoots of these varieties. To assess the potential of ancient Turkish hulled wheat species for saline agriculture, we investigated their antioxidant responses to varying salt stress levels. We also explored the role of exogenous GB in mitigating salt stress. Enzymatic and non-enzymatic antioxidant defense mechanisms were examined in both the roots and shoots of these varieties.

## 2. Results

### 2.1. Effects of Salt Stress on Plant Growth

#### 2.1.1. Effect of Salt Stress on the Fresh Weight of Plant Roots and Shoots

Under salt stress, hulled wheat varieties exhibited a decrease in root fresh weight as salt concentration increased (Table 1 and Appendix A, and Figure 1d). Significant reductions were observed at 200 mM NaCl and 200 mM KCl (30.41% and 30.21%, respectively). However, interestingly, root biomass also increased under salt stress, with the highest increase (13%) observed at 100 mM NaCl + 100 mM KCl + 0.5 mM GB (Table 2). This was followed by 50 mM NaCl + 50 mM KCl + 0.5 mM GB (8.8%), 30 mM KCl (5.8%), and 30 mM NaCl (3.98%). While individual applications of NaCl and KCl generally reduced the root fresh weight, combined applications mitigated this effect by approximately 50%. Additionally, GB supplementation further reduced the decrease by about 50%, and when combined with KCl, the reduction was approximately 70%. In contrast to the roots, the shoot fresh weight increased with salt concentrations up to 150 mM (Appendix A). Significant decreases were observed at 200 mM NaCl and 200 mM KCl (14.63% and 10.02%, respectively) (Table 1). Combined salt stress with GB supplementation reduced these decreases by 80% for NaCl and 54% for KCl. Furthermore, combined salt stress with GB support increased the shoot fresh weight by approximately 20% compared to the control. Among the varieties, *T. dicoccum* exhibited the highest fresh weight (199.83 g), followed by *T. monococcum* (196.59 g), *T. boeoticum* (147.17 g), and *T. speltoides* (141.58 g) (Appendix A).

#### 2.1.2. Effect of Salt Stress on the Dry Weight of Plant Roots and Shoots

Due to increased salt accumulation in the roots, the decrease in stem dry weight was more pronounced than in the roots, reaching approximately 94% at 200 mM salt. However, combined salt stress with GB supplementation mitigated this decrease by approximately 68% (Table 2 and Appendix A). While the stem dry weight increased by 2–44% with increasing salt doses (except for 200 mM), it decreased by 5.65% and 6.76% at 200 mM NaCl and 200 mM KCl, respectively. The highest shoot dry weight (44.37%) was observed at 50 mM NaCl + 50 mM KCl + 0.5 mM GB (Table 2). Among the varieties, *T. dicoccum* exhibited the highest dry weight (26.71 g), followed by *T. monococcum* (24.91 g), *T. boeoticum* (20.67 g), and *T. speltoides* (18.99 g) (Appendix A).

#### 2.1.3. Effect of Salt Stress on Leaf Length

Salt doses up to 100 mM increased leaf length by approximately 9%, while higher doses reduced it by 5–17%. However, GB supplementation with high doses of NaCl and KCl (150 and 200 mM) increased leaf length, reaching 22.38% at 150 mM KCl + 0.5 mM GB, and 21.11% at 100 mM NaCl + 100 mM KCl + 0.5 mM GB (Table 1 and Table 2). Among the varieties, *T. dicoccum* exhibited the highest leaf length (23.28 cm), followed by *T. monococcum* (20.96 cm), *T. boeoticum* (20.26 cm), and *T. speltoides* (19.34 cm) (Table 3). The changes in leaf length are detailed in Appendix A and Figure 2b.

#### 2.1.4. Effect of Salt Stress on Total Plant Height

Changes in plant height paralleled those in leaf length. Generally, salt applications up to 150 mM increased plant height by 3–23%, while 150 and 200 mM applications reduced it (200 mM NaCl, 5.64%; 200 mM KCl, 12.91%) (Table 1, Appendix A). Combined salt stress with GB increased plant height by approximately 30%. *T. dicoccum* exhibited the highest total plant height (32.94 cm), followed by *T. monococcum* (30.44 cm), *T. boeoticum* (23.94 cm), and *T. speltoides* (22.12 cm) (Table 3). The changes in plant height are detailed in Appendix A.

#### 2.1.5. Effect of Salt Stress on Chlorophyll (Chl) a, b, and Total Chlorophyll and Carotene Content

At 30 and 50 mM NaCl and KCl, the Chl_a content increased by approximately 35%. However, higher salt doses reduced the Chl_a content, reaching a 45% decrease at 200 mM. GB supplementation mitigated this decrease by 10% compared to the control. Combined salt stress with GB increased the Chl_a content by approximately 30% compared to the control. In contrast, the Chl_b content increased by 1–15% with increasing NaCl doses. In KCl applications, this increase occurred up to 100 mM, and higher doses reduced the Chl_b content by 6–16%. Combined salt stress with GB increased the Chl_b content by approximately 16% compared to the control (Table 2, Appendix A, and Figure 1c). The chlorophyll a/b ratios decreased by 20–40% at 100 mM salt and above. A 30% increase was observed at 50 mM KCl. While 33% increases were observed at 50 mM NaCl + 50 mM KCl, and 50 mM NaCl + 50 mM KCl + 0.5 mM GB, no change was observed at 100 mM NaCl + 100 mM KCl + 0.5 mM GB compared to the control (Table 1, Table 2, Appendix A). The total chlorophyll content increased by 20–30% up to 100 mM salt but decreased by 10–30% at higher doses, both in individual and combined salt applications. GB-supplemented combined salt applications increased the total chlorophyll content by 20–30% (Table 2, Appendix A). The carotene content generally increased with increasing salt doses. This increase reached 100% in combined salt applications and 255% in GB-supplemented combined salt applications (Table 2, Appendix A). Among the varieties, *T. boeoticum* exhibited the highest Chl_a, Chl_b, total chlorophyll, and carotene content (2.26, 0.96, 3.24, and 0.66 mg g^−1^ FW, respectively), followed by *T. speltoides* (2.24, 0.94, 3.18, and 0.62 mg g^−1^ FW, respectively), *T. monococcum* (2.22, 0.90, 3.12, and 0.42 mg g^−1^ FW, respectively), and *T. dicoccum* (2.18, 0.90, 3.10, and 0.38 mg g^−1^ FW, respectively) (Table 3 and Appendix A). The highest chlorophyll a/b ratio was observed in *T. monococcum* (2.47) (Figure 2c).

### 2.2. Effect of Salt Stress on Soluble Protein Content

Compared to the control, the wheat varieties exhibited varying levels of protein content increases depending on salt doses: 3–24% for NaCl, 29–55% for KCl, 55–68% for combined salt, 60–90% for GB-supplemented salt, and 90–111% for GB-supplemented combined salt (Table 4 and Table 5).

### 2.3. Effects of Salt Stress on Antioxidant Enzyme Activity in Plant Roots and Shoots

#### 2.3.1. Effect on SOD Activity

Considering the whole plant, SOD activity increased the most in *T. monococcum* (19.30%) compared to the control. The lowest increase was observed in *T. dicoccum* (0.36%) (Table 6, Figure 2a and Figure 3a,b). SOD activity increased with increasing salt doses (Table 3). Root SOD activity was significantly higher than shoot activity (average SOD activity: root, 0.985 ± 0.14 U mg^−1^ FW; shoot, 0.867 ± 0.11 U mg^−1^ FW, *p* ≤ 0.01) (Table 4 and Table 5). The highest SOD activity in both roots and shoots was observed at 50 mM NaCl + 50 mM KCl + 0.5 mM GB (1.252 ± 0.16 U mg^−1^ FW in the root, 1.058 ± 0.10 U mg^−1^ FW in the shoot).

#### 2.3.2. Effect on CAT Activity

The catalase activity in hulled wheat increased between 112% and 231% compared to the control. The highest activity was observed in *T. boeoticum* and *T. speltoides* (231% and 228%, respectively), and the lowest in *T. monococcum* (112.9%) (Table 6, Figure 3c,d). The catalase activity was higher than the control in both roots and shoots at all salt doses (Table 4, Table 5, and Appendix A). The highest activity was observed in the shoots (0.050 ± 0.09 U mg^−1^ FW) at 30 mM NaCl, and in the roots (0.061 ± 0.11 U mg^−1^ FW) at 200 mM NaCl + 0.5 mM GB (Table 4 and Table 5).

#### 2.3.3. Effect on GR Activity

GR activity in hulled wheat showed a similar profile to the catalase activity. The highest increase compared to the control was observed in *T. boeoticum* (195.58%), and the lowest in *T. monococcum* (76.2%) (Table 6 and Figure 3e,f). While GR activity generally increased at various rates depending on salt doses, a slight decrease was observed in the shoot at 200 mM KCl, and 100 mM NaCl + 100 mM KCl (8.7% and 3.26%, respectively) (Table 4). The highest GR activity in both the shoots and roots was observed at 50 mM NaCl + 50 mM KCl + 0.5 mM GB (0.260 ± 0.07 U mg^−1^ FW in the roots; 0.213 ± 0.06 U mg^−1^ FW in the shoots) (Table 4 and Table 5). GR activity changes in the roots and shoots are given in Appendix A.

#### 2.3.4. Effect on GST Activity

*T. boeoticum*, which exhibited the highest SOD, CAT, and GR activity, had the lowest GST activity (0.97%) (Table 6, Figure 3g,h). Among the hulled wheat, the highest GST activity was observed in *T. dicoccum* (115.6%). The highest root GST activity was observed at 150 mM NaCl (0.181 ± 0.09 U mg^−1^ FW), while it decreased sharply at 200 mM NaCl (0.080 ± 0.01 U mg^−1^ FW) (Table 3). In the shoots, the highest GST activity was observed at 50 mM NaCl and 50 mM NaCl + 50 mM KCl + 0.5 mM GB (0.150 ± 0.02 U mg^−1^ FW and 0.149 ± 0.02 U mg^−1^ FW, respectively) (Table 4 and Table 5). GST activity changes in the roots and shoots are given in Appendix A.

#### 2.3.5. Effect on APX Activity

Ascorbate peroxidase (APX) showed the highest increase in activity in *T. dicoccum* (33.96%) under salt stress among hulled wheat (Table 6 and Figure 3i,j). *T. speltoides* had the least increase in APX activity (19.70%). The highest root APX accumulation was observed at 150 mM KCl (0.431 ± 0.05 U mg^−1^ FW), while the lowest was observed at 200 mM NaCl and 200 mM KCl (0.271 ± 0.01 U mg^−1^ FW and 0.276 ± 0.03 U mg^−1^ FW, respectively). In the shoot, the highest APX accumulation was observed at 50 mM NaCl + 50 mM KCl + 0.5 mM GB (0.416 ± 0.03 U mg^−1^ FW), while the lowest was observed at 200 mM NaCl and 200 mM KCl (0.277 ± 0.02 U mg^−1^ FW and 0.279 ± 0.02 U mg^−1^ FW, respectively) (Table 4 and Table 5). APX activity changes in the roots and shoots are given in Table 6 and Appendix A.

### 2.4. Effect on Proline Accumulation

Proline accumulation increased significantly in hulled wheat due to salt stress. The highest percentage increase compared to the control was observed in *T. monococcum* (1099.85%), and the lowest in *T. speltoides* (842.15%) (Table 6). The highest root proline accumulation was observed at GB-supplemented combined 50 and 100 mM NaCl and KCl applications (748.42 ± 54.23 U mg^−1^ FW and 772.67 ± 42.89 U mg^−1^ FW, respectively), while the lowest was observed at 200 mM NaCl and KCl (144.92 ± 13.40 U mg^−1^ FW and 141.08 ± 8.21 U mg^−1^ FW, respectively) (Table 4 and Table 5). Similarly, the highest shoot proline accumulation was observed at GB-supplemented 50 mM NaCl + 50 mM KCl (741.92 ± 45.99 U mg^−1^ FW) and 50 mM NaCl + 50 mM KCl (606.33 ± 32.36 U mg^−1^ FW). The lowest shoot proline accumulation was observed at 200 mM KCl and 200 mM NaCl (117.67 ± 8.27 U mg^−1^ FW and 117.75 ± 29.58 U mg^−1^ FW, respectively) (Table 4 and Table 5, and Figure 2d). Proline accumulation changes in the roots and shoots are given in Table 6 and Appendix A.

### 2.5. Effect on Lipid Peroxidation (LPO) (MDA)

LPO content increased approximately 18 times in hulled wheat compared to the control. The lowest increase was observed in *T. monococcum* (1751.75%), and the highest in *T. boeoticum* (1885.45%) (Table 6). The highest root LPO accumulation was caused by 200 mM KCl and 200 mM NaCl (385.08 ± 26.96 nmol g^−1^ FW and 379.17 ± 23.74 nmol g^−1^ FW, respectively). The lowest LPO accumulation in roots was observed at 50 mM KCl and GB-supplemented combined 50 mM salt applications (40.42 ± 3.52 nmol g^−1^ FW and 40.58 ± 1.83 nmol g^−1^ FW, respectively) (Table 4 and Table 5). The highest shoot LPO accumulation was again caused by 200 mM salt applications (NaCl, 394.67 ± 23.06 nmol g^−1^ FW and KCl, 369.08 ± 22.36 nmol g^−1^ FW). The least shoot LPO accumulation was observed at GB-supplemented 50 mM salt application, as in the roots (27.83 ± 1.75 nmol g^−1^ FW) (Table 4 and Table 5). MDA accumulation changes in the roots and shoots are given in Table 6 and Appendix A, and Figure 1a and Figure 2d.

### 2.6. ANOVA, MANOVA, and Levene Test Results

The statistical analysis results (between-subjects effects test, Levene’s test, and MANOVA) are presented in Appendix A. A GT biplot analysis explained 66.1% of the variation in *T. boeoticum* shoots. Features with high discriminatory power included MDA, DW/FW, DW, TP, LH, GST, APX, PRO, total Chl, Chl_a, FW, and Chl a/b. Conversely, SOD, CAT, and Chl_b had low discriminatory power. The *T. boeoticum* stems exhibited positive correlations between GR, PRO, and APX, as well as Chl_b, SOD, DW, PT, Carotene, PH, and LH. A negative correlation was observed between MDA, Chl a/b, and FW. Treatments T14 (200 mM NaCl + 0.5 mM GB) and T13 (150 mM NaCl + 0.5 mM GB) were prominent in terms of DF/FW. T18 (100 mM NaCl + 100 mM KCl + 0.5 mM GB) was prominent in SOD and Chl_b, T17 (50 mM NaCl + 50 mM KCl + 0.5 mM GB) in GST, PRO, and GR, T11 (50 mM NaCl + 50 mM KCl) in Chl a/b and FW, and T5 (200 mM NaCl) in MDA content. In the *T. boeoticum* roots, the GT biplot explained 72% of the variation. A negative correlation was observed between MDA, DW, and DW/FW, and a positive correlation was observed between APX, GP, and CAT. In the *T. dicoccum* shoots, the GT biplot analysis explained 75% of the variation. Positive correlations were observed between carotene and DW/FW, DW and GST, GR, CAT, LH, and PRO, SOD and APX, and Chl a/b and carotene. In the *T. monococcum* shoots, the GT biplot analysis explained 73.1% of the variation. CAT had low discriminatory power; carotene, LH, FW, and Chl_b had medium discriminatory power; and the other features had high discriminatory power. In the *T. speltoides* stems, the GT biplot analysis explained 70.6% of the variation. SOD, Chl_b, LH, and carotene had medium discriminatory power, while the other features had high discriminatory power. The control treatment was located far from the other treatments and the examined features (Figure 4).

Multiple comparison analyses revealed no significant difference between *T. speltoides* and *T. boeoticum* in terms of protein concentrations; between *T. monococcum* and *T. dicoccum* and *T. speltoides* and *T. boeoticum* in terms of CAT; between *T. monococcum* and *T. dicoccum* in terms of GR, and between *T. dicoccum* and *T. boeoticum* in terms of GST enzyme activities (*p* ≤ 0.174, *p* ≤ 0.971, *p* ≤ 0.886, *p* ≤ 0.679 and *p* ≤ 0.404, respectively) (Table 7). Similarly, no significant differences were found between the roots and shoots in terms of protein concentration (*p* ≤ 0.098), SOD (*p* ≤ 0.254), and GST (*p* ≤ 0.193) enzyme activities. The antioxidant responses of wheat to different salt doses did not show significant differences in CAT (*p* ≤ 0.165) and GST (*p* ≤ 0.310) enzyme activities. Additionally, there were no significant differences in CAT (*p* ≤ 0.420) and GST (*p* ≤ 0.532) activities between plant parts under different salt doses. When all the variables were considered together (Wheats × Sections × Doses), no significant differences were observed in CAT (*p* ≤ 0.414), GR (*p* ≤ 0.131), and GST (*p* ≤ 0.383) enzyme activities (Appendix A).

Levene’s test is a statistical test used to evaluate the homogeneity of data. It tests the null hypothesis that the error variance of the dependent variable is equal across groups. The result of the Levene test allows us to determine whether the difference between groups is statistically significant. If the difference is significant as a result of the Levene test, it means that there is an inequality of variance between the groups, and in this case, we may need to use different analysis methods. In our study, the *p* values of all the tested variables were greater than 0.05, which suggests that there was no variance inequality between the groups and that no other testing tool was needed. Levene’s test indicated significant differences in the activities of all the antioxidant enzymes (*p* ≤ 0.001) (Appendix A).

The MANOVA multivariate test creates four test statistics (Pillai’s Trace, Wilks’ Lambda, Hotelling’s Trace, and Roy’s Largest Root), and if *p* ≥ 0.05 in the tested factors, these test statistics are looked at. In this study, although the F values for each test statistic changed, the null hypothesis of MANOVA was rejected because *p* ≤ 0.05 (*p* ≤ 0.001), and it was concluded that both wheat varieties, plant parts, and salt stress doses were important in the enzymatic/non-enzymatic antioxidative response (Appendix A). This test revealed significant differences in enzyme activities across all applications (Wheats × Sections × Doses) (Pillai’s Trace, V: 2.566, F: 2.65, *p* ≤ 0.001; Wilks’ Lambda, V: 0.030, F: 3.040, *p* ≤ 0.001; Hotelling’s Trace, V: 5.237, F: 3.567, *p* ≤ 0.001; and Roy’s Largest Root, V: 2.272, F: 12.746, *p* ≤ 0.001) (Appendix A).

A GT biplot analysis can be considered reliable if it explains more than 50% of the variation [26]. In this study, the explained variation exceeded this threshold (>65%). Biplot analysis is a versatile technique that can be applied to various genotypes by inputting bidirectional data (genotype–trait). This method allows for the screening of genotypes based on desired traits [27] and visually represents the relationships between examined features. According to the biplot analyses, the following antioxidant enzymes were found to be at the forefront of combating salt stress in the roots of hulled wheat: APX, GR, GST, and SOD in *T. monococcum*; SOD and CAT in *T. dicoccum*; and only SOD in *T. speltoides* and *T. boeoticum*. In the shoots, the response pattern differed: GR, GST, and SOD in *T. monococcum*; SOD and APX in *T. dicoccum*; GR in *T. speltoides*; and GST in *T. boeoticum* were the primary responders. Additionally, GB-supported combined salt applications enhanced the activities of antioxidant enzymes.

## 3. Discussion

### 3.1. Effects of Salt Stress on Plant Growth

#### 3.1.1. Effect on Plant Biomass

Salt stress induces ion toxicity, disrupts nutritional balance, and deteriorates physiological processes, leading to significant reductions in plant yield [28]. Moreover, salt stress triggers oxidative stress by disrupting enzymatic activities, photosynthesis, membrane integrity, ionic homeostasis, hormonal balance, and water and nutrient uptake [12,29]. Guo et al. [8] and Zou et al. [9] observed decreased root and shoot lengths and dry weight in wheat under 100 mM salt stress. In our study, hulled wheat varieties exposed to salt stress exhibited a decrease in root fresh weight with increasing salt doses (5–30%). Significant reductions were observed at 200 mM NaCl or KCl (30.41% and 30.21%, respectively). Excessive Na^+^, K^+^, and Cl^−^ ions hinder the uptake of essential nutrients, altering plant processes. Guo et al. [8] reported decreased K+, Ca^2+^, and Zn^2+^ uptake and increased Na^+^ and Cl^−^ uptake in salt-sensitive wheat. The observed decrease in fresh and dry weight with increasing salt concentrations indicates that high levels of sodium, potassium, and chloride ions disrupt ion balance, leading to nutrient deficiencies and cellular water loss.

Fortmeier and Schubert [30] reported that high sodium concentrations interfere with K^+^ accumulation and stomatal regulation. However, increasing Na^+^ ion concentration in vacuoles via the tonoplast pathway driven by the proton gradient is a critical strategy for salt tolerance. Neubert [31] reported that plants have evolved mechanisms to protect essential organelles, such as the cytosol, from excess sodium. In our study, while individual applications of 200 mM NaCl or KCl caused approximately 90% weight loss, the combined application of 100 mM NaCl + 100 mM KCl resulted in only a 5% weight loss. This suggests that the type of salt stress (sodium-based or potassium-based) influences plant development. Plants may be more tolerant to combined stress when K^+^ can be simultaneously taken up to maintain intracellular K^+^/Na^+^ balance. Therefore, applying potassium-based fertilizers to plants grown in sodium-rich soils could mitigate salt stress.

Plants accumulate Na^+^ ions in root vacuoles using the tonoplast pathway to reduce sodium transport to the stem and leaves [31]. By optimizing K^+^ uptake, plants not only restrict Na^+^ entry but also benefit from sodium removal from the cell under salt stress. Wakeel [32] reported that this mechanism helps maintain the K^+^/Na^+^ ratio in the cytosol, ensuring plant survival under salinity conditions. In our study, the increase in shoot dry and fresh weight by approximately 20% and plant height by 30%, with the combined application of 100 mM NaCl + 100 mM KCl, suggests that sodium and potassium ions are transported to the shoot rather than being stored in the root. Due to increased salt accumulation in the roots, shoot biomass decreases significantly, reaching approximately 94% at 200 mM salt. Although 200 mM salt application reduces plant height, biomass loss can be compensated by approximately 68% with glycine betaine supplementation, and plant height can increase by 30% compared to the control.

Salt stress is a polygenic trait controlled by multiple genes. Na^+^ exclusion, K^+^ uptake, maintenance of the optimal K^+^/Na^+^ ratio, osmotic regulation, and antioxidant enzyme regulation are crucial for salt tolerance [10]. To enhance salt tolerance in plants, various techniques are employed, such as the screening and selection of tolerant genotypes, genetic engineering, and agronomic practices, but these can be costly and time-consuming [11]. In contrast, strategies such as applying osmoprotectants (e.g., glycine betaine and proline), seed coating, nutrient management, and hormone application (auxin, gibberellic acid, and brassinosteroids) can offer promising results [33]. In this study, individual applications of sodium or potassium salts, especially at 100 mM, significantly stressed all hulled wheat varieties, inhibiting plant growth. However, glycine betaine supplementation mitigated salt stress, especially when combined with sodium and potassium chloride. This suggests that applying glycine betaine as an osmoprotectant and potassium-based fertilizers can improve wheat cultivation in sodic/saline soils.

#### 3.1.2. Effect on Chlorophyll a, b, and Total Chlorophyll and Carotene Content

Photosynthesis, a critical process for plant survival, can be hindered by salt stress [34]. Ion accumulation (Na^+^, K^+^, and Cl^−^) in chloroplasts and decreased plant water potential due to high salt stress can block photosynthesis [35]. Guo et al. [8] observed that salinity stress leads to stomatal closure, reduced CO_2_ absorption, and decreased transpiration rates in wheat. Additionally, high salt stress significantly reduces photosynthetic pigments in chloroplasts, which in turn, significantly reduces photosynthetic efficiency and productivity. In our study, we observed a 45% decrease in the Chl_a content at 200 mM salt doses. However, when 0.5 mM GB was added as an osmoprotectant, the Chl_a content improved by 10%. Furthermore, combined salt stress with GB increased the Chl_a content by 30% compared to individual salt stress or the control. This demonstrates that exogenous GB significantly enhances the photosynthetic activity of hulled wheat.

Salinity stress can lead to ionic toxicity, decreased leaf growth, reduced carboxylation, decreased photosynthesis, and premature leaf abscission. Additionally, it reduces the efficiency of PS-II, stomatal conductance, intercellular CO_2_, and electron transport, all of which contribute to decreased photosynthesis [7]. Sarker and Oba [36] stated that NaCl or KCl disrupts the K^+^/Na^+^ ion balance, especially in plant root cells, leading to ion accumulation in the stem and leaves. These ions can trigger the production of ROS through the active energy released during photosynthesis, damaging photosystems I and II. Damage to PS-I reduces the Chl_a content, altering the Chl a/b ratio and reducing PS-II efficiency. High salt concentrations can also decrease the number of stomata and increase stomatal closure, further inhibiting CO_2_ absorption and photosynthesis [37,38]. In our study, 150 and 200 mM individual salt applications significantly decreased the chl_a content and the chlorophyll a/b ratio, indicating damage to photosystem I. However, combined salt stress, especially with GB supplementation, appeared to restore photosynthetic activity, suggesting that maintaining intracellular K^+^/Na^+^ ion balance is crucial for photosynthetic efficiency.

Salt stress can also affect chloroplast ultrastructure. It significantly inhibits PS-II, which is essential for light energy conversion and photosynthetic efficiency [39]. In wheat, granum thylakoids of chloroplasts become loosely arranged under 200 mM NaCl [40]. Increasing the number of chloroplasts is a strategy used by halophytes to cope with salinity stress [41]. The significant decrease (16–20%) in Chl_b content at 150–200 mM salt doses indicates severe damage to PS-II in hulled wheat. However, combined salt stress with GB increased the Chl_b content by 16%, suggesting a protective role of GB in chlorophyll biosynthesis. Insufficient energy during salt stress reduces molecular oxygen and increases ROS production, including H_2_O_2_, O_2_^−^, ^1^O_2_, and OH^•^ [42,43]. Plant cells must constantly resist oxidative damage, and excessive ROS production can be harmful. Therefore, ROS production must be regulated to maintain redox homeostasis [44,45]. Our study suggests that exogenous GB with potassium ions significantly contributes to ROS control.

The carotene accumulation increased with increasing salt stress. Carotene, a non-enzymatic antioxidant, protects photosystems from ROS damage. The carotene content increased up to 100% with GB-supplemented salt applications and up to 150% with combined GB-supplemented salt applications. *T. boeoticum*, one of the oldest wheat ancestors, exhibited the highest Chl_a, Chl_b, total Chl, and carotene content, suggesting its resilience to salt stress. Our study indicates that hulled wheat can maintain photosynthetic activity up to 100 mM salt stress. However, at higher salt concentrations, photosynthetic activity decreases due to damage to photosystems. Combined salt stress with GB can maintain photosynthetic activity and even increase the total chlorophyll content by 20–30%.

### 3.2. Effect of Salt Stress on Soluble Protein Concentrations

To survive ionic, oxidative, and osmotic stress, plants produce various osmoprotectants (such as proline, glycine betaine, dimethylsulfoniopropionate (DMSP), and trehalose) and other unidentified proteins. Both long- and short-term osmotic response pathways trigger the biosynthesis and accumulation of compatible osmolytes that stabilize proteins, cellular structures, and morphology, restoring cellular osmotic potential. El-Sayed [46] previously reported that salt-tolerant bean varieties have lower protein content but higher proline and amino acid levels compared to salt-sensitive varieties. Compatible osmolytes have been shown to prevent water loss during short-term osmotic stress and increase cellular turgor and expansion during long-term stress. Notably, many osmolytes synthesized under salt stress are also accumulated under drought and cold stress, with their biosynthesis being partially species- and tissue-specific [1].

Several studies have demonstrated that the exogenous application of osmoprotectants can alleviate salt and metal stress in plants [14,47]. In this study, we observed a 3–111% increase in soluble protein content depending on increasing salt stress doses. Notably, the approximately 111% increase in soluble protein content in GB-supplemented combined salt applications is significant. This suggests that both combined sodium and potassium chloride application (60–90% protein increase) and exogenous GB application (90–111% protein increase) trigger the production of both enzymatic and non-enzymatic antioxidants in hulled wheat, leading to an effective defense against ROS.

### 3.3. Effects of Salt Stress on the Enzymatic and Non-Enzymatic Antioxidant Defense System

Plants possess an intricate antioxidant defense system comprising enzymatic and non-enzymatic antioxidants that scavenge ROS within cellular organelles. If ROS production surpasses the scavenging capacity of this system, oxidative damage ensues. The enzymatic components of this system include SOD, CAT, APX, GR, GST, monodehydroascorbate reductase (MDHAR), dehydroascorbate reductase (DHAR), glutathione peroxidase (GPX), and peroxiredoxin (PRX). Non-enzymatic components encompass ascorbate (AsA), glutathione (GSH), carotenoids, alkaloids, tocopherols, flavonoids, non-protein amino acids, and phenolic compounds [8,9,11].

The AsA-GSH cycle, involving AsA, GSH, and four antioxidant enzymes (APX, DHAR, GR, and MDHAR), plays a pivotal role in regulating ROS homeostasis by detoxifying H_2_O_2_ [48]. Carotenoids, flavonoids, and phenolic acids contribute to ROS homeostasis regulation by scavenging free radicals [49,50]. These components collectively facilitate plant adaptation to oxidative stress induced by salt stress and mediate stress signal transduction by controlling ROS homeostasis. Figure 5 illustrates the antioxidant defense system activated in plants under salt stress. While the SOS pathway is implicated in both potassium and sodium uptake and is crucial for K^+^/Na^+^ homeostasis regulation in plants [51], the precise mechanisms of active potassium uptake regulation, including direct regulation, remain elusive.

Sodium ions are initially sensed by sensors localized in the plasma membrane (PM). Salt stress induces ionic stress, which results in changes in the calcium status of the cytosol [1]. Glycosyl inositol phosphorylceramides (GIPCs) are abundant in the PM and receive these signals [52]. SOS2 kinase is induced by sodium [53], while salt stress induces ROS stress [1], which in turn modulates the transcription level of SOS1 [54]. By interacting with CAT, SOS2 connects the SOS pathway to other signaling pathways and phosphorylates and activates SOS1 [55]. On the other hand, GIPC increases calcium signaling by binding to Na^+^, while calcium receptors bind to intracellular Ca^2+^ and activate Na^+^/H^+^ antiporter activity. MPK6, activated by phosphatidic acid (PA), phosphorylates SOS1 and increases its activity. FER senses changes in the cell wall under salt stress and mediates calcium signaling for long-term stress. ANNEXINs (ANNs) modulate calcium signaling under salt stress, promoting the activation of SOS2 activity by SCaBP8. ROS released under stress activate the enzymatic and non-enzymatic antioxidant defense system. Many osmoregulators, carotenes, alkaloids, flavonoids, tocopherols, phenolic compounds, non-protein amino acids, and a number of yet unidentified metabolites, are activated to support antioxidant defense. The SOS pathway and the released ROS force gene regulatory systems to come into play. Following all these processes, the ROS released in the organism are neutralized, and ionic homeostasis and subsequently cellular stress resistance are achieved [51].

Numerous studies have demonstrated the role of the antioxidant defense system in mitigating oxidative damage during biotic and abiotic stress in plants [12,29,57,58,59]. Meneguzzo et al. [60] reported a strong correlation between antioxidants and salinity tolerance in wheat species. Sreenivasulu et al. [61] observed increased antioxidant enzyme activities under salt stress. Athar et al. [62] reported decreased K^+^/Na^+^ ratios in sensitive and tolerant wheat varieties under high salt stress (150 mM NaCl), accompanied by reduced growth and photosynthetic activity. However, tolerant varieties exhibited increased endogenous AsA production and CAT activity to counteract salt stress. Furthermore, salt-tolerant plants have been shown to resist salinity by releasing sodium ions from leaves and enhancing SOD, APX, and CAT activities, as well as photosynthetic activity through AsA [63]. In our study, SOD activity increased in hulled wheat varieties with increasing salt doses, with significantly higher activity in roots compared to shoots (0.867 ± 0.11 units mg^−1^ FW in the shoots, 0.985 ± 0.14 units mg^−1^ FW in the roots). The highest SOD activity increase in both the roots and shoots was observed at 50 mM NaCl + 50 mM KCl + 0.5 mM GB. Among the varieties, *T. boeoticum* exhibited the highest SOD activity increase (18.27%), while *T. dicoccum* showed the lowest (8.17%). The 15% increase in SOD activity in *T. boeoticum* and *T. monococcum* under 150 mM salt stress suggests their superior tolerance to high salinity compared to other varieties. In these species, SOD activity increased by 75% in glycine betaine-supplemented combined salt applications compared to the control.

Tetraploid wheat is generally more sensitive to salt than bread wheat [64], likely due to lower K^+^ accumulation in the leaves, which is controlled by the Kna1 loci on chromosome 4D [65]. In our study, *T. dicoccum* exhibited lower activities of all enzymatic and non-enzymatic antioxidants. HKT genes are implicated in sodium ion exclusion under salinity stress. Yang et al. [66] reported that TaHKT1;5-D alters transcriptional programming in *Aegilops tauschii* under salt stress, while Byrt et al. [67] observed no change in hexaploid wheat. In contrast, Zhao et al. [68] and Wang et al. [69] reported the reduced function of this gene in the JN177 hexaploid wheat variety. These conflicting results raise questions about the tissue specificity and HKT gene-based regulation of TaHKT1;5-D. In our study, the data for *T. speltoides*, a hexaploid wheat, aligns with Byrt et al. [67] in terms of its antioxidant enzyme activities under salt stress.

Jabeen et al. [56] observed increased H_2_O_2_ and O_2_^−^ concentrations in wheat under 300 mM salt stress, accompanied by increased SOD, POD, CAT, and proline accumulations to enhance salt tolerance (276 mg g^−1^ DW, 7.05 mg g^−1^ DW, 7.60 mg g^−1^ DW, and 65.69 mg g^−1^ DW, respectively). Zeeshan et al. [70] found that 100 mM salt stress induced ROS and MDA accumulation, leading to significant upregulation of POD, CAT, APX, and GR activities to mitigate salt stress effects. Dong et al. [71] reported that 120 mM salinity stress induced oxidative and osmotic stresses, with significant upregulation of POD, SOD, and CAT activities to reduce salt-induced damage. Ahanger et al. [72] found that 100 mM salt stress caused hydrogen peroxide and superoxide accumulation, leading to significant upregulation of CAT, SOD, and APX activities to scavenge ROS. Mandhania et al. [73] reported that 10 dSm^−1^ salt stress induced ROS accumulation and increased MDA content, while CAT and APX activities were upregulated to counteract oxidative stress. In our study, CAT activity increased by 112–231%, SOD by 1–19%, GR by 76–196%, GST by 1–116%, and APX by 20–34% in hulled wheat under salt stress compared to the control. *T. boeoticum* and *T. speltoides* exhibited the highest increases in SOD, CAT, and GR activities. Interestingly, *T. dicoccum* showed the highest GST and APX activities, despite lower levels of other antioxidant enzymes. This suggests that GST and APX are the primary antioxidant defense enzymes in *T. dicoccum*. In *T. boeoticum*, CAT and GR activities increased up to 300% in combined salt applications and glycine betaine-supplemented combined salt applications, suggesting an active role of HKT genes in maintaining K^+^/Na^+^ ion balance, as proposed by Yang et al. [66], Zhao et al. [68], and Wang et al. [69].

### 3.4. Effects of Salt Stress on Proline Accumulation

When plants encounter osmotic stress, they employ osmoregulation, accumulating sugars, polyols, amino acids, and quaternary ammonium compounds to mitigate the adverse effects of stress [13]. Osmoregulation plays a crucial role in triggering the defense mechanism against reactive oxygen species (ROS) to regulate plant–water relations [57]. Proline, an osmoprotectant, aids in osmotic adjustment, ROS detoxification, and the strengthening of the PS-II structure [74]. Previous studies have reported that various antioxidant enzymes (SOD, APX, and CAT) increase their activities with exogenously applied glycine betaine (GB) and significantly improve the salinity tolerance of wheat [75]. In our study, we observed a 2–10-fold increase in proline accumulation in hulled wheat due to increasing salt doses. Another study reported that GB application (10 and 30 mM) increased germination, calcium, and chlorophyll content in shoots and leaves, enhancing the salinity tolerance of wheat (*Triticum aestivum* L.) [76]. Similarly, exogenous proline application (60 ppm) was found to down-regulate malondialdehyde (MDA) and improve salinity tolerance in wheat [77].

Rao et al. [16] reported that increased proline and GB production reduced the harmful effects of salt stress by activating antioxidant enzymes. Exogenous osmoprotectant applications have also been shown to increase proline and potassium accumulation, improve the K^+^/Na^+^ ratio, and stabilize protein and lipid structures [78]. Consequently, hormone and osmoprotectant applications enhance antioxidant activities, photosynthetic efficiency, and membrane stability, facilitating significant recovery under salinity stress by detoxifying ROS. In our study, the highest proline accumulation was observed in the 50 mM NaCl + 50 mM KCl + 0.5 mM GB treatment. Notably, when 50 and 100 mM combined sodium and potassium chloride applications were supplemented with 0.5 mM GB, hulled wheat exhibited a 10–300% increase in all antioxidant enzymes, especially SOD, CAT, and APX. These findings provide evidence that proline accumulation increases under salt stress. Furthermore, exogenous GB application supports this proline increase and significantly aids the plant in managing ROS induced by salt stress.

### 3.5. Effects of Salt Stress on Lipid Peroxidation (LPO; MDA)

Hasanuzzaman et al. [79] observed that salt-sensitive wheat varieties subjected to salinity stress exhibited higher levels of H_2_O_2_ and lipid peroxidation compared to salt-tolerant varieties. Zou et al. [9] reported that malondialdehyde (MDA) levels in wheat seedlings exposed to 100 mM NaCl for 5 and 10 days increased by 35% and 68%, respectively. In our study, MDA accumulation increased 2-fold at 30 mM NaCl, 3–4-fold at 50 mM, 15–17-fold at 100 mM, 27–29-fold at 150 mM, and 39–42-fold at 200 mM NaCl, compared to the control. No significant difference was observed between NaCl and KCl applications in terms of MDA accumulation (*p* ≥ 0.784; Appendix A). However, combined NaCl and KCl applications (50 mM NaCl + 50 mM KCl; total 100 mM) resulted in approximately 5-fold lower MDA accumulation compared to individual applications. This suggests that membrane damage induced by salt stress is influenced by salt type rather than concentration, and that damage is significantly reduced when the K^+^/Na^+^ balance can be maintained.

Previous studies have indicated that ionic homeostasis, a key process regulating ion flow to maintain low Na^+^ and high K^+^ concentrations, plays a crucial role [13,80]. Intracellular Na^+^ and K^+^ ion regulation (homeostasis) depends on various cytosolic enzymes and maintains membrane potential and cell volume. To maintain balanced Na^+^ and K^+^ concentrations in the cytosol, plants excrete excess salt through primary and secondary active transport and accumulate these ions in plasma and tonoplast membranes to preserve homeostasis during salt stress [80]. Cordovilla et al. [81] reported the up- and down-regulation of various K^+^ genes in response to salt stress. The vacuole also compartmentalizes to protect the cytosol from excessive Na^+^ ions [13]. Previous studies have demonstrated that plants utilize affinity-based transporters found in biological membranes (related to the K^+^/Na^+^ balance) for K^+^ uptake [82,83].

While global food security is threatened by increasing salinization, current EU policies, such as the Common Agricultural Policy and the European Green Deal, do not adequately address this issue [84,85]. There is a pressing need for the EU to develop specific policies and regulations to mitigate the effects of salt-affected soils and promote sustainable agriculture in these challenging environments [86]. By prioritizing research, policy development, and sustainable agricultural practices, we can safeguard global food security and ensure a resilient future for wheat production.

## 4. Materials and Methods

### 4.1. Plant Cultivation and Stress Management Practices

Four ancient hulled wheat species native to Türkiye (CGN 06600: *T. monococcum* L.; VIR 21007: *T. boeoticum* Boiss; CGN 10684: *T. dicoccum* Schrank; TRI 17070: *T. speltoides*) obtained from various Gene Banks (VIR, The N.I. Vavilov—Russian Scientific Research Institute of Plant Industry, Saint Petersburg, Russia; TRI, IPK—Gatersleben, Germany; CGN, Centre For Genetic Resources, Wageningen, The Netherlands) were grown in culture containers (15 cm wide × 6 cm deep) under controlled conditions (25 °C, 70% humidity, 14/10 light/dark cycle). Seeds (20 seeds in each culture pot) were sterilized, sown, and grown for 10 days without stress in Hoagland solution (pH 6.8; EC: 3.1 dS m^−1^) [87]. Subsequently, salt stress was applied for 15 days. Before each salt application, the old salt solution was poured out and the roots of the plants were cleaned of excess solution with blotting paper and 50 mL of salt solution [NaCl (30–50–100–150–200 mM), KCl (30–50–100–150–200 mM), NaCl + KCl (50 mM NaCl + 50 mM KCl and 100 mM NaCl + 100 mM KCl), and combinations with 0.5 mM glycine betaine] applied daily. Control plants received only Hoagland solution. After the stress period, plant roots and shoots were harvested, frozen in liquid nitrogen, and stored at −20 °C. All experiments were conducted in triplicate.

### 4.2. Measuring the Effects of Salinity on Plant Growth Parameters: Fresh Weight, Dry Weight, Leaf Length, and Height

After a 15-day stress period, plants were harvested, and the fresh and dry weights of roots and shoots were determined. Root and shoot lengths were also measured. Prior to fresh weight measurement, roots were washed with deionized water to remove salts. Samples were dried at 65 °C for 48 h to obtain dry weight.

### 4.3. Preparation of Crude Enzyme Extracts

In crude enzyme extract (except APX), the method of Yilmaz et al. [88] was used with minor modifications. A tissue sample (0.5 g) was crushed using a pre-chilled mortar and pestle on ice in 2 mL of 0.1 M potassium phosphate buffer (pH 7.5) containing 1% polyvinyl polypyrrolidone (PVPP), 0.1% EDTA, and 15,000 g. The supernatant was collected by centrifugation at +4 °C for 20 min and labeled as crude enzyme extract. For APX, 0.5 g of fresh tissue was ground in liquid nitrogen and suspended in 2 mL of buffer consisting of 50 mM Tris-HCl (pH 7.2), 2% PVP, 1 mM Na_2_EDTA, and 2 mM ascorbate [89]. Crude enzyme extracts were stored at −20 °C until use. Soluble protein (SP) concentrations were determined spectrophotometrically at 595 nm using the Bradford [90] method. Bovine serum albumin (BSA) fraction V was used as the standard. The results were recorded as mg mL^−1^ protein.

### 4.4. Determination of the Chlorophyll and Carotene Content

Chlorophyll and carotene content were determined using the method described by Yilmaz et al. [91]. Briefly, 100 mg of fresh leaf tissue was extracted with dimethyl sulfoxide (DMSO, 7 mL) at 65 °C until complete decolorization. The extract was diluted to 10 mL, and absorbance was measured at 647, 663, and 470 nm (in UV-1800 Shimadzu Spectrophotometer, Shimadzu Corporation, Kyoto, Japan). Chlorophyll and carotene concentrations were calculated using standard formulas and expressed as mg g^−1^ fresh weight.(1)$$Chl\;a(\frac{mg}{gr}fw)=\left(12.25\times A663\right)-(2.79\times A647)$$(2)$$Chl\;b(\frac{mg}{gr}fw)=\left(21.50\times A647\right)-(5.1\times A663)$$(3)$$Total\;Chl(\frac{mg}{gr}fw)=\left(7.15\times A663\right)+(18.71\times A647)$$(4)$$Carotene(\frac{mg}{gr}fw)=\frac{\left(1000\times A470\right)-\left(1.82-Chl\;a\right)-(85.02\times Chl\;b)}{198}$$

### 4.5. Determination of Enzyme Activity

#### 4.5.1. Catalase (CAT) (EC 1.11.1.6) Activity

Catalase activity was measured using the method of Duman et al. [78] with minor modifications. Briefly, the decrease in absorbance at 25 °C was monitored for 3 min at 240 nm in a reaction mixture containing 20 mM sodium hydrogen phosphate (NaHPO_4_) buffer (pH 7.5), 15 mM hydrogen peroxide (H_2_O_2_), and 50 µL crude enzyme extract. The molar absorption coefficient of H_2_O_2_ is 40 mM^−1^ cm^−1^. Specific activity (SA) was calculated as described by Yilmaz et al. [91] and expressed as units per milligram of protein.(5)$$SA\;\left(\frac{unite}{mg}protein\right)=\frac{\frac{∆Abs}{min}}{40}\times \frac{crude\;enzyme\;vol}{cuvette\;vol}\times \frac{1}{prot\;cons}\times 1000$$

#### 4.5.2. Superoxide Dismutase (SOD) (EC 1.15.1.1) Activity

Superoxide dismutase (SOD) activity was determined using a modified method of Duman et al. [78]. A reaction mixture containing sodium phosphate buffer (NaHPO_4_, pH 7.4), EDTA (0.1 mM), methionine (10 mM), NBT (0.1 mM), and riboflavin (0.005 mM) was prepared in a light-proof amber bottle. Standard SOD solutions (10–500 ng mL^−1^) and crude enzyme extracts (20 µL) were added to the reaction mixture. Samples and standards were exposed to light (150 µmol m^−2^ s^−1^, distance of 20 cm from the lam) for 15 min, and absorbance was measured at 560 nm. SOD activity was calculated based on the percentage inhibition of NBT reduction, with one unit of SOD defined as the amount of enzyme required to inhibit 50% of NBT reduction. All experiments were performed in triplicate.(6)$$Inh\%=\frac{Light\;Control\;Abs.-Sample\;Abs}{Light\;Control\;Abs.}\times 100$$

Enzyme concentrations were plotted against percent inhibition on a logarithmic scale. A standard curve of log SOD concentration versus percent inhibition was generated. Sample SOD concentrations were determined from this curve and expressed as units per milligram of protein.

#### 4.5.3. Ascorbate Peroxidase (APX) (EC 1.11.1.11) Activity

APX activity was determined following the method of Akbulut and Çakır [89], with minor modifications. A reaction mixture containing 50 mM potassium phosphate buffer (pH 7.0), 0.20 mM ascorbate, 10 mM H_2_O_2_, and 50 μL of crude enzyme extract was prepared in a final volume of 1 mL. The reaction was initiated by adding H_2_O_2_, and the decrease in ascorbate concentration was monitored at 290 nm for 3 min. Buffer served as a blank. The reaction was conducted in a quartz cuvette at 25 °C. The specific activity of APX was calculated based on the consumption of H_2_O_2_, using the extinction coefficient (ε) of H_2_O_2_ (2.8 mM^−1^ cm^−1^ at 290 nm), as described by Akbulut and Çakır [89]. Results were expressed as units per milligram of protein.(7)$$SA\;\left(\frac{unite}{mg}protein\right)=\frac{\frac{∆Abs}{min}}{2.8}\times \frac{crude\;enzyme\;vol}{cuvette\;vol}\times \frac{1}{prot\;cons}\times 1000$$

#### 4.5.4. Glutathione Reductase (GR) (EC 1.6.4.2) Activity

GR activity was measured according to the method described by Misra and Gupta [92]. A reaction mixture containing 100 mM potassium phosphate buffer (KHPO_4_; pH 7.5), 0.1 mM EDTA, 0.1 mM nicotine amide adenine dinucleotide phosphate (NADPH), 1 mM oxidized glutathione (GSSG), and 50 µL of crude enzyme extract was incubated at 25 °C for 5 min. The decrease in absorbance at 340 nm was monitored, and the buffer served as a blank. The specific activity of GR was calculated using the molar absorption coefficient (ε) of NADPH at 340 nm (6.2 mM^−1^ cm^−1^) and expressed as units per milligram of protein.(8)$$SA\;\left(\frac{unite}{mg}protein\right)=\frac{\frac{∆Abs}{min}}{6.2}\times \frac{crude\;enzyme\;vol}{cuvette\;vol}\times \frac{1}{prot\;cons}\times 1000$$

#### 4.5.5. Glutathione S-Transferase (GST) (EC 2.5.1.18) Activity

GST activity was measured following the protocol of Yilmaz et al. [91]. The reaction mixture, buffered with 100 mM potassium phosphate (pH 7.5), contained 0.1 mM EDTA, 0.1 mM NADPH, 1 mM glutathione (GSH), and 1 mM 1-chloro-2,4-dinitrobenzene (CDNB). After adding 50 µL of crude enzyme extract, the reaction was incubated for 5 min to allow for non-specific activity. Subsequently, the change in absorbance at 340 nm was monitored for 5 min in a 2 mL quartz cuvette at 25 °C. Given that the molar absorption coefficient (ε) of NADPH at 340 nm is 6.2 mM^−1^ cm^−1^, the specific activity of the samples was calculated and expressed as units per milligram of protein.(9)$$SA\;\left(\frac{unite}{mg}protein\right)=\frac{\frac{∆Abs}{min}}{6.2}\times \frac{crude\;enzyme\;vol}{cuvette\;vol}\times \frac{1}{prot\;cons}\times 1000$$

### 4.6. Lipid Peroxidation (LPO, MDA) Assessment

To evaluate the impact of salt stress on lipid peroxidation, the thiobarbituric acid reactive substances (TBARS) content in the tissues was measured. TBARS are formed as byproducts of lipid peroxidation, a process initiated by reactive oxygen species (ROS). While direct measurement of ROS is challenging due to their short half-life, TBARS analysis provides an indirect assessment of oxidative damage to cell membranes [93]. This method quantifies malondialdehyde (MDA), a primary product of lipid peroxidation [94]. The lipid peroxidation levels were determined using the method of Madhava and Sresty [95], with minor modifications. Fresh tissue (0.5 g) was homogenized in 5 mL of 0.1% trichloroacetic acid (TCA) and centrifuged (12,000× *g* for 5 min). The supernatant was mixed with 20% TCA containing 0.5% 2-thiobarbituric acid (TBA) (for every 1 mL of supernatant, 4 mL of 20% TCA solution) and incubated in boiling water for 30 min. After centrifugation, the absorbance was measured at 532 nm and 600 nm (background correction) using a spectrophotometer. The TBARS content was calculated using the molar absorption coefficient of MDA (155 mM^−1^ cm^−1^) and expressed as nmol g^−1^ Fw.

### 4.7. Determination of Proline Content

Proline content was determined following the method of Temizgul et al. [96]. Fresh tissue (0.25 g) was ground to a fine powder in liquid nitrogen using a pre-cooled mortar and pestle. The powder was homogenized in 5 mL of 3% sulphosalicylic acid and centrifuged at 5000× *g* for 10 min at 4 °C. Two milliliters of the supernatant were transferred to screw-cap glass tubes and mixed with 2 mL of acid ninhydrin. The mixture was vortexed immediately and then supplemented with 2 mL of 96% acetic acid and 1 mL of 3% sulphosalicylic acid. After another round of vortexing, the tubes were incubated in a boiling water bath for 1 h. Subsequently, the tubes were rapidly cooled in a water bath and 4 mL of toluene was added. The mixture was vortexed vigorously to ensure thorough mixing. The tubes were then left in the dark for 1–2 h to allow the toluene to extract the colored product. Absorbance was measured at 520 nm against toluene as a blank. A standard curve was constructed using 10 proline standards ranging from 0.01 µM to 1.5 mM. The proline content of the samples was calculated from the standard curve and expressed as nmol g^−1^ Fw.

### 4.8. Statistical Analysis

To evaluate the impact of salt stress on different wheat varieties and plant parts, a comprehensive statistical analysis was conducted using two-way and three-way analysis of variance (ANOVA). Two-way ANOVA, implemented in SPSS 28.0, assessed the main effects and interactions between salt doses and wheat varieties, as well as salt doses and plant parts. A three-way ANOVA, performed using GraphPad Prism version 10.0.0, examined the combined effects of salt doses, plant parts (root and shoot), and wheat varieties (*T. monococcum*, *T. dicoccum*, *T. speltoides*, and *T. boeoticum* ). To further delineate significant differences between treatment groups, multiple comparison tests were employed, including Duncan’s multiple range test, the least significant difference (LSD) test, and Tukey’s test. All statistical analyses were conducted at a significance level of *p* ≤ 0.05. To visualize the relationships between genotypes and traits, genotype biplot analysis was performed using GenStat 12.0 statistical software. These biplots provided a clear representation of which genotypes exhibited superior performance for specific traits [97]. Each experiment was replicated three times, and data are presented as mean ± standard deviation (SD). Different lowercase letters (a, b, c, d, e) within tables and figures indicate statistically significant differences between means.

## 5. Conclusions

Our research indicates that potassium fertilization is more beneficial than nitrogenous fertilization in saline soils. Potassium enhances plant salt tolerance and normalizes biological activities. When potassium and sodium salts are applied together, the plant’s tolerance to salt stress is higher than when potassium or sodium salts are applied individually. However, exogenous glycine betaine application protects the plant from the destructive effects of salt stress to a significant extent. These results suggest that exogenous glycine betaine addition together with potassium-based fertilizers in wheat farming in saline/sodic soils may be beneficial in increasing plant resistance and ultimately crop yield.

By reintroducing salt-tolerant genes from wild relatives and ancient wheat species into breeding programs, we can develop more resilient crops. Beyond genetic approaches, the exogenous applications of osmoprotectants, phytohormones, seed treatments, and nutrient management can further enhance salt tolerance in wheat. These strategies, combined with genetic improvements, can mitigate the negative impacts of salinity stress and boost wheat productivity. Cultivating salt-tolerant ancient wheat in saline areas could contribute to both food security and the demand for healthier, more sustainable food options. In particular, it is beneficial to study the genetic structure of the *T. boeoticum* variety related to salinity with advanced molecular techniques and include this variety in breeding programs to develop new salt-resistant varieties.

## Figures and Tables

**Figure 1 plants-14-00678-f001:**
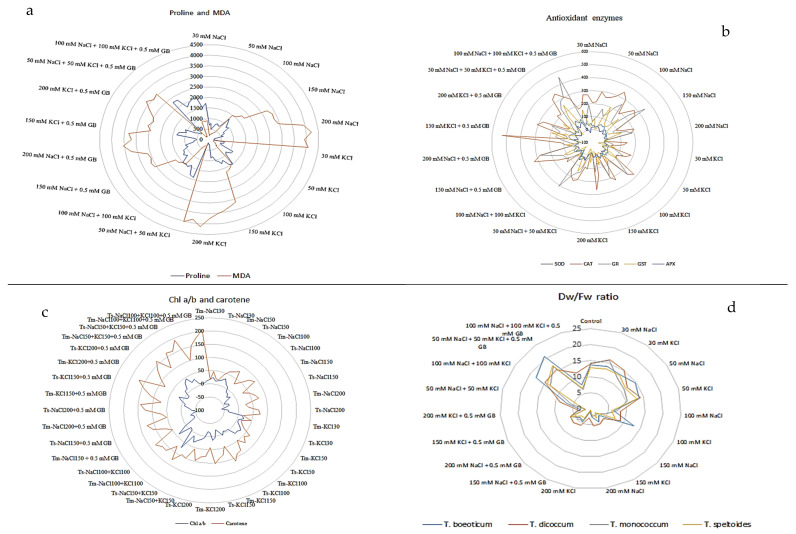
Percentage change of traits/compounds under salt stress compared to control. (**a**) Proline and MDA; (**b**) Antioxidant enzymes; (**c**) Chl a/b and carotene; (**d**) Dry weight (Dw)/Fresh weight (Fw). Tm: *T. monococcum*; Td: *T. dicoccum*; Tb: *T. boeoticum*; Ts: *T. speltoides.* (**a**,**b**,**d**) are given to show the averages of the data obtained from four wheat varieties.

**Figure 2 plants-14-00678-f002:**
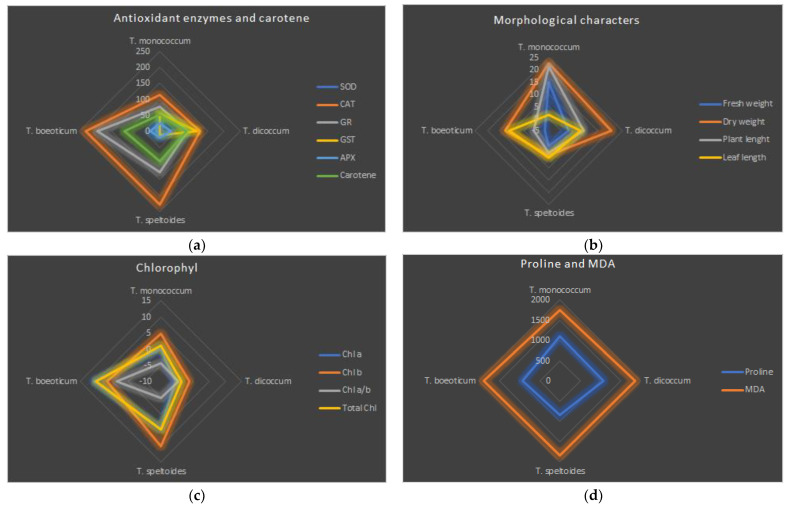
Individual responses of hulled wheat to salt treatments. (**a**) Percentage change of antioxidant enzymes and carotene under salt stress compared to control. (**b**) Percentage change of morphological characters under salt stress compared to control. (**c**) Percentage change of chlorophyll under salt stress compared to control. (**d**) Percentage change of proline and MDA under salt stress compared to control.

**Figure 3 plants-14-00678-f003:**
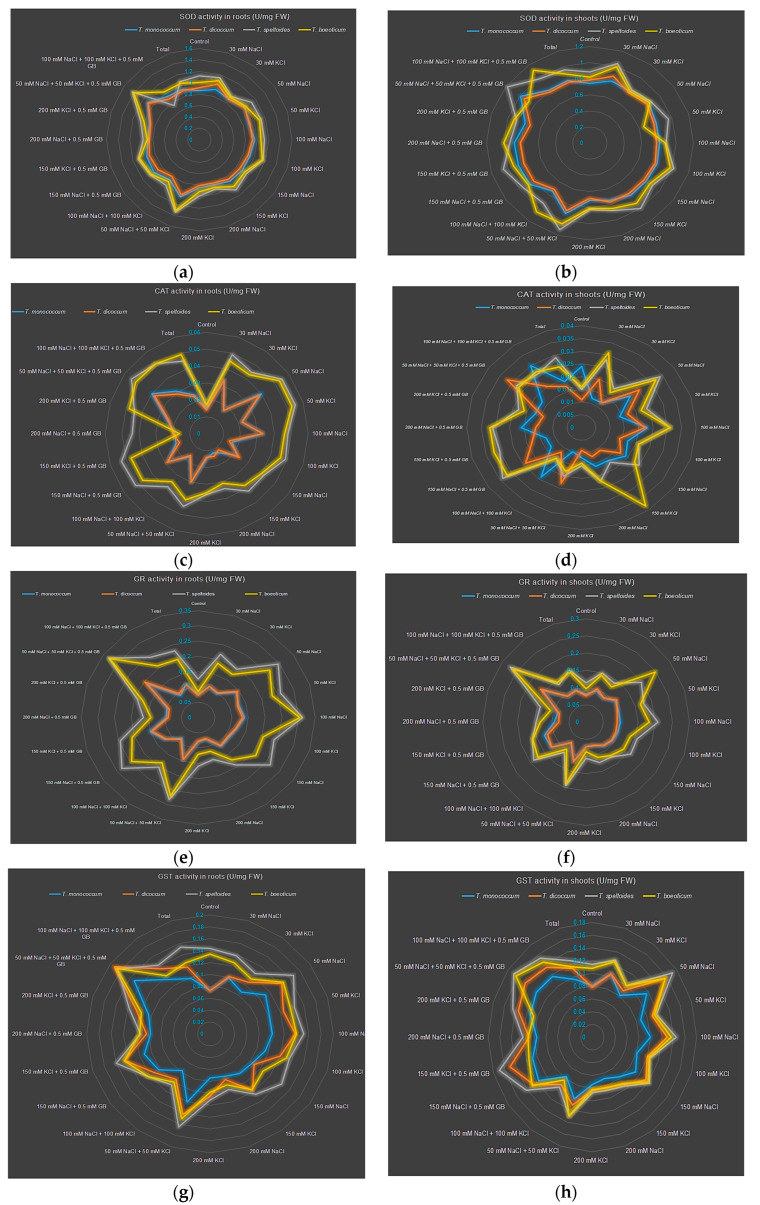
Antioxidant defense responses of hulled wheats under salt treatments. (**a**,**b**) SOD activity in root and shoot, (**c**,**d**) CAT activity in root and shoot, (**e**,**f**) GR activity in root and shoot, (**g**,**h**) GST activity in root and shoot, (**i**,**j**) APX activity in root and shoot.

**Figure 4 plants-14-00678-f004:**
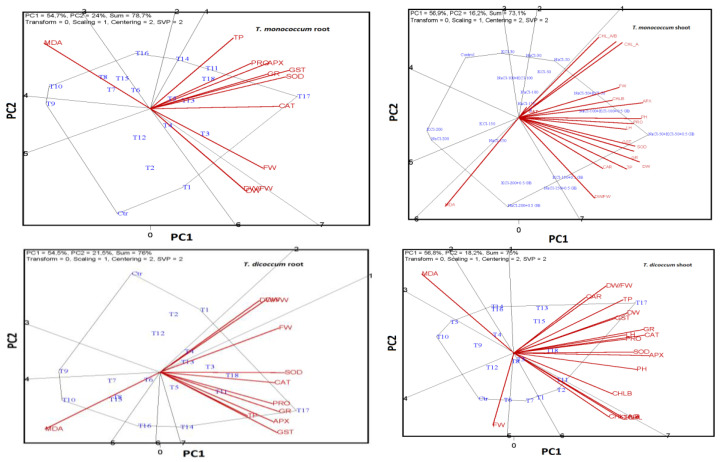
Polygon views of the GT biplot based on symmetrical scaling for the which-won-what pattern for traits. Biplot plot showing the interactions between all components of the “Wheats × Root × Salt Stress” factors. Ctr: Control; T1: 30 mM NaCl; T2: 50 mM NaCl; T3: 100 mM NaCl; T4: 150 mM NaCl; T5: 200 mM NaCl; T6: 30 mM KCl; T7: 50 mM KCl; T8: 100 mM KCl; T9: 150 mM KCl; T10: 200 mM KCl; T11: 50 mM NaCl + 50 mM KCl; T12: 100 mM NaCl + 100 mM KCl; T13: 150 mM NaCl + 0.5 mM GB; T14: 200 mM NaCl + 0.5 mM GB; T15: 150 mM KCl + 0.5 mM GB; T16: 200 mM KCl + 0.5 mM GB; T17: 50 mM NaCl + 50 mM KCl + 0.5 mM GB; T18: 100 mM NaCl + 100 mM KCl + 0.5 mM GB.

**Figure 5 plants-14-00678-f005:**
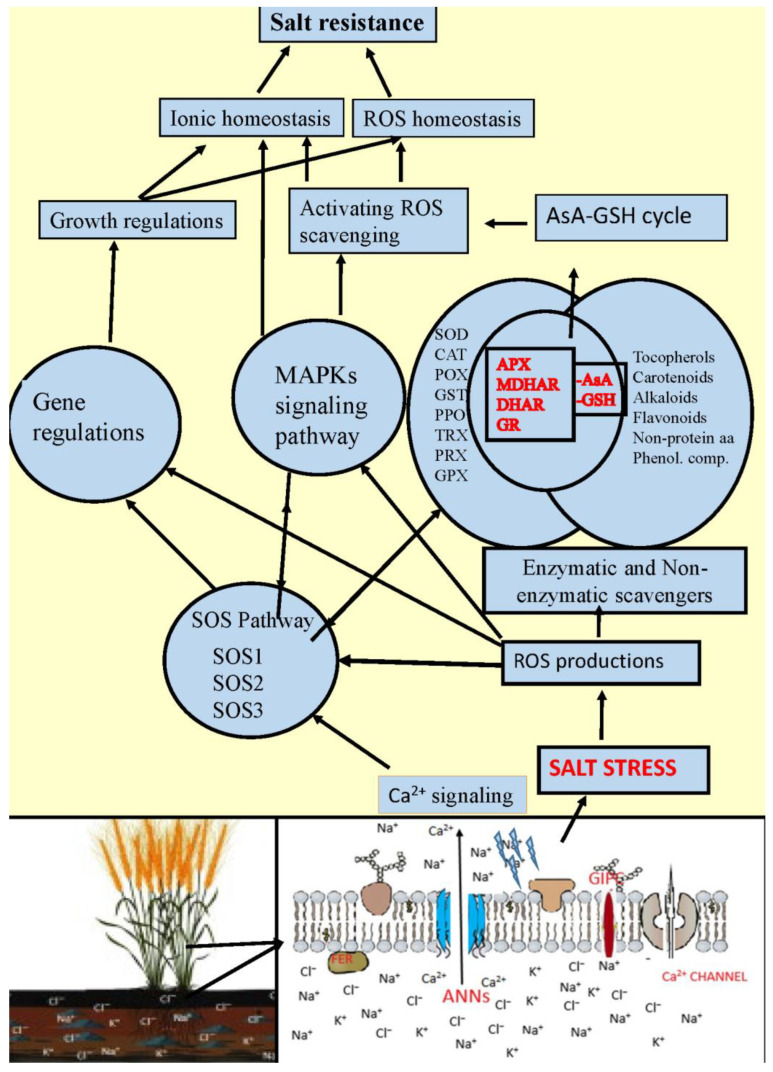
Schematic representation of the antioxidative defense system developed under salt stress in hulled wheats. (In the schematic representation of the antioxidative defense response that develops under salt stress in hulled wheat, data presented in the studies of Yang and Guo [1], Zhu [51], Jiang et al. [52], Lin et al. [53], Chung et al. [54], Verslues et al. [55], and Jabeen et al. [56] were used).

**Table 1 plants-14-00678-t001:** Effect of salt applications on the growth of plant shoots in hulled wheat.

Salt Treatments	Section	Fresh Weight (g)	Dry Weight (g)	Dw/Fw (%)	Plant Height (cm)	Leaf Height (cm)	Chl a (mg g^−1^ fw)	Chl b (mg g^−1^ fw)	Chl a/b	Total Chl (mg g^−1^ fw)	Carotene (mg g^−1^ fw)
*Control*	*S*hoot	29.28 ± 0.55 ^c^	3.68 ± 0.09 ^b^	12.57 ^ab^	25.17 ± 0.47 ^bc^	19.75 ± 0.33 ^b^	2.18 ± 0.03 ^bc^	0.89 ± 0.01 ^b^	2.45 ^c^	3.08 ± 0.03 ^bc^	0.29 ± 0.03 ^a^
*30 mM NaCl*	*S*hoot	32.11 ± 0.09 ^d^	4.32 ± 0.09 ^c^	13.45 ^c^	29.75 ± 0.45 ^e^	20.50 ± 0.33 ^c^	2.77 ± 0.02 ^c^	0.97 ± 0.01 ^c^	2.86 ^d^	3.74 ± 0.02 ^d^	0.36 ± 0.03 ^b^
*30 mM KCl*	*S*hoot	32.30 ± 0.19 ^d^	3.83 ± 0.20 ^b^	11.86 ^a^	27.25 ± 0.33 ^d^	21.42 ± 0.39 ^d^	2.51 ± 0.01 ^bc^	0.92 ± 0.01 ^b^	2.73 cd	3.43 ± 0.02 ^cd^	0.35 ± 0.03 ^ab^
*50 mM NaCl*	*S*hoot	33.15 ± 0.22 ^d^	4.52 ± 0.07 ^d^	13.64 ^c^	30.915 ± 0.41 ^e^	20.42 ± 0.39 ^c^	2.96 ± 0.01 ^c^	1.03 ± 0.01 ^c^	2.87 ^d^	3.99 ± 0.02 ^e^	0.46 ± 0.03 ^bc^
*50 mM KCl*	*S*hoot	33.35 ± 0.21 ^d^	4.20 ± 0.06 ^c^	12.59 ^b^	27.67 ± 0.33 ^d^	21.17 ± 0.33 ^d^	2.89 ± 0.01 ^c^	0.96 ± 0.02 ^c^	3.01 d	3.86 ± 0.02 ^de^	0.42 ± 0.05 ^b^
*100 mM NaCl*	*S*hoot	31.28 ± 0.30 ^cd^	4.28 ± 0.08 ^c^	13.68 ^c^	29.25 ± 0.33 ^e^	18.75 ± 0.33 ^b^	2.00 ± 0.01 ^b^	0.98 ± 0.01 ^c^	2.04 ^b^	2.98 ± 0.02 ^bc^	0.47 ± 0.03 ^bc^
*100 mM KCl*	*S*hoot	31.83 ± 0.35 ^cd^	4.33 ± 0.11 ^c^	13.60 ^c^	25.92 ± 0.33 ^c^	20.09 ± 0.33 ^c^	2.26 ± 0.02 ^bc^	0.96 ± 0.01 ^c^	2.35 bc	3.21 ± 0.02 ^c^	0.52 ± 0.03 ^c^
*150 mM NaCl*	*S*hoot	30.27 ± 0.23 ^c^	3.90 ± 0.10 ^bc^	12.88 ^b^	27.50 ± 0.33 ^d^	20.84 ± 0.33 ^d^	1.39 ± 0.02 ^b^	0.94 ± 0.01 ^b^	1.48 ^a^	2.41 ± 0.02 ^ab^	0.45 ± 0.03 ^bc^
*150 mM KCl*	*S*hoot	30.37 ± 0.27 ^c^	3.76 ± 0.09 ^b^	12.38 ^b^	24.09 ± 0.41 ^b^	17.42 ± 0.25 ^ab^	1.95 ± 0.01 ^b^	0.84 ± 0.01 ^a^	2.32 bc	2.80 ± 0.01 ^b^	0.51 ± 0.04 ^c^
*200 mM NaCl*	*S*hoot	24.99 ± 0.34 ^a^	3.48 ± 0.07 ^a^	13.93 ^c^	23.75 ± 0.33 ^b^	19.75 ± 0.33 ^b^	1.21 ± 0.02 ^a^	0.90 ± 0.02 ^b^	1.34 ^a^	2.11 ± 0.02 ^a^	0.44 ± 0.03 ^b^
*200 mM KCl*	*S*hoot	26.34 ± 0.36 ^b^	3.43 ± 0.09 ^a^	13.02 ^b^	21.92 ± 0.39 ^a^	16.33 ± 0.33 ^a^	1.76 ± 0.01 ^b^	0.75 ± 0.02 ^a^	2.35 ^bc^	2.51 ± 0.02 ^ab^	0.50 ± 0.05 ^c^
*50 mM NaCl + 50 mM KCl*	*S*hoot	33.20 ± 0.37 ^c^	4.38 ± 0.06 ^b^	13.19 ^ab^	28.42 ± 0.39 ^bc^	21.83 ± 0.39 ^ab^	2.87 ± 0.02 ^c^	0.88 ± 0.09 ^b^	3.26 ^c^	3.75 ± 0.11 ^c^	0.55 ± 0.02 ^b^
*100 mM NaCl + 100 mM KCl*	*S*hoot	30.77 ± 0.22 ^bc^	3.81 ± 0.13 ^a^	12.38 ^a^	23.49 ± 0.25 ^a^	19.17 ± 0.33 ^a^	2.32 ± 0.01 ^b^	0.90 ± 0.02 ^b^	2.58 ^b^	3.22 ± 0.03 ^b^	0.59 ± 0.02 ^b^
*150 mM NaCl + 0.5 mM GB*	*S*hoot	30.72 ± 0.41 ^bc^	4.88 ± 0.12 ^bc^	15.89 ^b^	30.58 ± 0.45 ^c^	24.17 ± 0.39 ^bc^	1.49 ± 0.02 ^a^	0.97 ± 0.01 ^bc^	1.54 ^a^	2.47 ± 0.02 ^a^	0.55 ± 0.02 ^b^
*200 mM NaCl + 0.5 mM GB*	*S*hoot	25.78 ± 0.45 ^a^	4.45 ± 0.26 ^b^	17.26 ^c^	28.75 ± 0.33 ^bc^	22.67 ± 0.45 ^b^	1.38 ± 0.01 ^a^	0.95 ± 0.01 ^bc^	1.45 ^a^	2.33 ± 0.02 ^a^	0.57 ± 0.02 ^b^
*150 mM KCl + 0.5 mM GB*	*S*hoot	31.13 ± 0.61 ^c^	4.72 ± 0.30 ^bc^	15.16 ^b^	27.67 ± 0.45 ^b^	23.00 ± 0.39 ^b^	2.17 ± 0.01 ^b^	0.89 ± 0.01 ^b^	2.44 ^b^	3.06 ± 0.01 ^b^	0.65 ± 0.02 ^bc^
*200 mM KCl + 0.5 mM GB*	*S*hoot	26.96 ± 0.68 ^ab^	4.38 ± 0.15 ^b^	16.25 ^bc^	26.25 ± 0.45 ^b^	22.34 ± 0.41 ^b^	1.96 ± 0.01 ^b^	0.81 ± 0.01 ^a^	2.42 ^b^	2.77 ± 0.01 ^ab^	0.64 ± 0.02 ^bc^
*50 mM NaCl + 50 mM KCl + 0.5 mM GB*	*S*hoot	28.69 ± 0.55 ^b^	5.32 ± 0.23 ^c^	18.54 ^c^	32.75 ± 0.45 ^c^	25.42 ± 0.51 ^c^	3.37 ± 0.01 ^d^	1.04 ± 0.01 ^c^	3.24 ^c^	4.41 ± 0.02 ^d^	0.73 ± 0.02 ^c^
*100 mM NaCl + 100 mM KCl + 0.5 mM GB*	*S*hoot	31.27 ± 0.76 ^c^	4.73 ± 0.17 ^bc^	15.13 ^b^	28.92 ± 0.45 ^c^	23.92 ± 0.53 ^bc^	2.72 ± 0.01 ^c^	1.03 ± 0.01 ^c^	2.64 ^b^	3.77 ± 0.01 ^c^	0.74 ± 0.02 ^c^

Means with different letters in the columns are significantly different at *p* < 0.05 level. The roots and shoots have been evaluated independently of each other. Values are averages of four *Triticum* species. Fw: Fresh weight; Dw: Dry weight.

**Table 2 plants-14-00678-t002:** Effect of salt applications on the growth of plant roots in hulled wheat.

Salt Treatments	Section	Fresh Weight (g)	Dry Weight (g)	Dw/Fw (%)
*Control*	Root	4.17 ± 0.26 ^c^	0.58 ± 0.07 ^c^	13.91 ^c^
*30 mM NaCl*	Root	4.33 ± 0.11 ^c^	0.65 ± 0.06 ^d^	15.01 ^d^
*30 mM KCl*	Root	4.41 ± 0.06 ^c^	0.59 ± 0.06 ^c^	13.38 ^c^
*50 mM NaCl*	Root	3.96 ± 0.12 ^bc^	0.48 ± 0.03 ^bc^	12.12 ^c^
*50 mM KCl*	Root	4.08 ± 0.09 ^c^	0.53 ± 0.06 ^c^	12.99 ^c^
*100 mM NaCl*	Root	3.63 ± 0.09 ^b^	0.26 ± 0.03 ^b^	7.16 ^b^
*100 mM KCl*	Root	3.64 ± 0.05 ^b^	0.30 ± 0.06 ^b^	8.24 ^b^
*150 mM NaCl*	Root	3.32 ± 0.10 ^b^	0.10 ± 0.01 ^a^	3.01 ^a^
*150 mM KCl*	Root	3.34 ± 0.09 ^b^	0.13 ± 0.02 ^ab^	3.89 ^a^
*200 mM NaCl*	Root	2.90 ± 0.05 ^a^	0.10 ± 0.02 ^a^	3.45 ^a^
*200 mM KCl*	Root	2.91 ± 0.07 ^a^	0.04 ± 0.01 ^a^	1.38 ^a^
*50 mM NaCl + 50 mM KCl*	Root	3.88 ± 0.09 ^b^	0.25 ± 0.04 ^b^	6.44 ^c^
*100 mM NaCl + 100 mM KCl*	Root	3.59 ± 0.09 ^b^	0.54 ± 0.09 ^cd^	15.04 ^de^
*150 mM NaCl + 0.5 mM GB*	Root	3.76 ± 0.13 ^b^	0.17 ± 0.02 ^ab^	4.52 ^b^
*200 mM NaCl + 0.5 mM GB*	Root	3.21 ± 0.15 ^a^	0.15 ± 0.02 ^ab^	4.67 ^b^
*150 mM KCl + 0.5 mM GB*	Root	4.18 ± 0.14 ^c^	0.24 ± 0.03 ^b^	5.74 ^bc^
*200 mM KCl + 0.5 mM GB*	Root	3.61 ± 0.14 ^b^	0.09 ± 0.02 ^a^	2.49 ^a^
*50 mM NaCl + 50 mM KCl + 0.5 mM GB*	Root	4.74 ± 0.23 ^d^	0.40 ± 0.02 ^c^	8.44 ^c^
*100 mM NaCl + 100 mM KCl + 0.5 mM GB*	Root	4.53 ± 0.28 ^d^	0.77 ± 0.03 ^d^	17.00 ^e^

Means with different letters in the columns are significantly different at *p* < 0.05 level. The roots and shoots have been evaluated independently of each other. Values are averages of four *Triticum* species. GB: Glycine betaine.

**Table 3 plants-14-00678-t003:** Effects of salt treatments on plant growth and antioxidant defense in hulled wheats.

Wheats	N	Soluble Protein (µg mL^−1^ Protein)	SOD (U mL^−1^ Protein)	CAT (U mL^−1^ Protein)	GR (U mL^−1^ Protein)	GST (U mL^−1^ Protein)	APX (U mL^−1^ Protein)	PRO (nmol g^−1^ fw)	MDA (nmol g^−1^ fw)
** *Control* **	12	383.25 ^a^	0.89 ^b^	0.01 ^a^	0.09 ^a^	0.10 ^a^	0.29 ^a^	66.31 ^a^	18.35 ^a^
** *T. monococcum* **	114	508.26 ^c^	0.873 ^b^	0.021 ^b^	0.106 ^b^	0.097 ^a^	0.320 ^b^	332.55 ^b^	166.19 ^b^
** *T. dicoccum* **	114	513.75 ^c^	0.847 ^a^	0.020 ^b^	0.104 ^b^	0.116 ^b^	0.299 ^a^	362.71 ^bc^	157.84 ^b^
** *T. speltoides* **	114	480.14 ^b^	1.015 ^d^	0.037 ^c^	0.194 ^c^	0.140 ^c^	0.348 ^c^	400.55 ^c^	194.34 ^c^
** *T. boeoticum* **	114	482.62 ^b^	0.971 ^c^	0.037 ^c^	0.177 ^c^	0.121 ^b^	0.333 ^b^	385.48 ^bc^	187.09 ^c^
** *Wheats* **	** *FW* ** ** *(g)* **	** *DW* ** ** *(g)* **	** *PH* ** ** *(cm)* **	** *LL* ** ** *(cm)* **	** *Chl a* ** *(mg g^−1^ fw)*	** *Chl b* ** *(mg g^−1^ fw)*	** *Chl a/b* **	** *Total Chl* ** *(mg g^−1^ fw)*	** *Carotene* ** *(mg g^−1^ fw)*
** *Control* **	16.72 ^b^	2.13 ^b^	25.17 ^b^	19.75 ^a^	2.18 ^a^	0.89 ^a^	2.45 ^bc^	3.08 ^c^	0.29 ^a^
** *T. monococcum* **	19.66 ^c^	2.49 ^c^	30.44 ^c^	20.96 ^b^	2.22 ^ab^	0.90 ^a^	2.47 ^c^	3.12 ^c^	0.42 ^c^
** *T. dicoccum* **	19.98 ^c^	2.67 ^d^	32.94 ^d^	23.28 ^c^	2.18 ^a^	0.90 ^a^	2.42 ^b^	3.10 ^c^	0.38 ^b^
** *T. speltoides* **	14.16 ^a^	1.90 ^a^	22.12 ^a^	19.34 ^a^	2.24 ^ab^	0.94 ^b^	2.38 ^ab^	3.18 ^b^	0.62 ^d^
** *T. boeoticum* **	14.72 ^a^	2.07 ^b^	23.94 ^a^	20.26 ^b^	2.26 ^b^	0.96 ^b^	2.35 ^a^	3.24 ^a^	0.66 ^e^

Means with different letters in the columns are significantly different at *p* < 0.05 level. FW: Fresh weight; DW: Dry weight; PH: Plant height; LL: Leaf length.

**Table 4 plants-14-00678-t004:** Effect of salt treatments on the enzymatic and non-enzymatic antioxidant defense system and soluble protein in hulled wheat shoots.

Salt Treatments	Section	Soluble Protein (µg mL^−1^ Protein)	SOD (U mL^−1^ Protein)	CAT (U mL^−1^ Protein)	GR (U mL^−1^ Protein)	GST (U mL^−1^ Protein)	APX (U mL^−1^ Protein)	PRO (nmol g^−1^ fw)	MDA (nmol g^−1^ fw)
*Control*	Shoot	450.17 ± 32.23 ^d^	0.808 ± 0.06 ^ab^	0.014 ± 0.01 ^a^	0.092 ± 0.02 ^a^	0.095 ± 0.02 ^a^	0.300 ± 0.02 ^b^	63.63 ± 10.43 ^a^	17.31 ± 1.14 ^a^
*30 mM NaCl*	Shoot	476.17 ± 38.04 ^de^	0.931 ± 0.10 ^d^	0.050 ± 0.09 ^e^	0.122 ± 0.03 ^b^	0.115 ± 0.01 ^b^	0.333 ± 0.02 ^d^	259.58 ± 30.37 ^c^	34.25 ± 5.21 ^a^
*30 mM KCl*	Shoot	467.17 ± 18.10 ^d^	0.795 ± 0.02 ^a^	0.016 ± 0.01 ^ab^	0.110 ± 0.03 ^ab^	0.093 ± 0.01 ^a^	0.304 ± 0.02 ^bc^	251.33 ± 22.55 ^c^	32.92 ± 1.78 ^a^
*50 mM NaCl*	Shoot	502.08 ± 32.38 ^e^	0.850 ± 0.02 ^b^	0.029 ± 0.01 ^bc^	0.190 ± 0.17 ^cd^	0.150 ± 0.02 ^c^	0.350 ± 0.02 ^e^	311.25 ± 41.93 ^d^	40.25 ± 6.18 ^a^
*50 mM KCl*	Shoot	521.92 ± 14.03 ^e^	0.802 ± 0.22 ^ab^	0.020 ± 0.01 ^b^	0.133 ± 0.04 ^b^	0.106 ± 0.01 ^ab^	0.325 ± 0.02 ^cd^	388.50 ± 22.04 ^e^	45.83 ± 8.87 ^a^
*100 mM NaCl*	Shoot	522.08 ± 27.56 ^e^	0.835 ± 0.05 ^b^	0.026 ± 0.01 ^bc^	0.145 ± 0.05 ^bc^	0.128 ± 0.02 ^bc^	0.327 ± 0.02 ^cd^	385.75 ± 33.30 ^e^	158.00 ± 25.17 ^bc^
*100 mM KCl*	Shoot	509.00 ± 20.04 ^e^	0.904 ± 0.11 ^cd^	0.018 ± 0.01 ^ab^	0.119 ± 0.03 ^b^	0.097 ± 0.01 ^ab^	0.322 ± 0.02 ^cd^	486.58 ± 22.94 ^f^	152.08 ± 18.80 ^b^
*150 mM NaCl*	Shoot	526.33 ± 28.24 ^e^	0.832 ± 0.05 ^b^	0.020 ± 0.01 ^b^	0.119 ± 0.03 ^b^	0.112 ± 0.01 ^b^	0.302 ± 0.03 ^bc^	279.75 ± 32.43 ^cd^	242.50 ± 82.30 ^c^
*150 mM KCl*	Shoot	473.08 ± 17.46 ^d^	0.874 ± 0.10 ^bc^	0.020 ± 0.02 ^b^	0.104 ± 0.02 ^ab^	0.089 ± 0.01 ^a^	0.307 ± 0.01 ^bc^	379.17 ± 23.05 ^de^	355.75 ± 27.34 ^d^
*200 mM NaCl*	Shoot	461.00 ± 27.28 ^d^	0.813 ± 0.06 ^ab^	0.017 ± 0.01 ^ab^	0.094 ± 0.02 ^a^	0.080 ± 0.01 ^a^	0.279 ± 0.02 ^a^	117.75 ± 29.58 ^ab^	394.67 ± 23.06 ^e^
*200 mM KCl*	Shoot	404.00 ± 25.08 ^c^	0.758 ± 0.06 ^a^	0.012 ± 0.01 ^a^	0.084 ± 0.01 ^a^	0.082 ± 0.01 ^a^	0.277 ± 0.02 ^a^	117.67 ± 8.27 ^ab^	369.08 ± 22.36 ^de^
*50 mM NaCl + 50 mM KCl*	Shoot	563.42 ± 15.92 ^f^	0.995 ± 0.11 ^ef^	0.021 ± 0.01 ^b^	0.155 ± 0.04 ^bc^	0.118 ± 0.01 ^b^	0.347 ± 0.02 ^de^	606.33 ± 32.26 ^gh^	47.75 ± 9.03 ^a^
*100 mM NaCl + 100 mM KCl*	Shoot	488.83 ± 22.67 ^de^	0.813 ± 0.10 ^ab^	0.014 ± 0.01 ^a^	0.089 ± 0.02 ^a^	0.087 ± 0.01 ^a^	0.304 ± 0.03 ^bc^	451.67 ± 49.49 ^ef^	157.67 ± 17.31 ^bc^
*150 mM NaCl + 0.5 mM GB*	Shoot	553.83 ± 21.20 ^f^	0.888 ± 0.06 ^c^	0.028 ± 0.01 ^bc^	0.153 ± 0.03 ^bc^	0.128 ± 0.01 ^bc^	0.326 ± 0.02 ^cd^	417.83 ± 34.86 ^e^	250.08 ± 13.07 ^c^
*200 mM NaCl + 0.5 mM GB*	Shoot	538.25 ± 15.20 ^ef^	0.887 ± 0.05 ^c^	0.023 ± 0.01 ^b^	0.113 ± 0.03 ^ab^	0.109 ± 0.01 ^ab^	0.291 ± 0.02 ^b^	247.50 ± 26.57 ^c^	325.42 ± 18.01 ^d^
*150 mM KCl + 0.5 mM GB*	Shoot	596.17 ± 6.71 f^g^	0.934 ± 0.10 ^d^	0.024 ± 0.01 ^b^	0.143 ± 0.02 ^bc^	0.136 ± 0.03 ^bc^	0.320 ± 0.03 ^cd^	437.83 ± 22.51 ^e^	256.33 ± 11.58 ^cd^
*200 mM KCl + 0.5 mM GB*	Shoot	608.42 ± 23.65 ^fg^	0.822 ± 0.07 ^b^	0.019 ± 0.01 ^ab^	0.105 ± 0.02 ^ab^	0.125 ± 0.02 ^bc^	0.296 ± 0.02 ^b^	210.33 ± 7.89 ^c^	285.00 ± 13.08 ^cd^
*50 mM NaCl + 50 mM KCl + 0.5 mM GB*	Shoot	661.33 ± 23.03 ^g^	1.058 ± 0.10 ^f^	0.029 ± 0.01 ^bc^	0.213 ± 0.06 ^d^	0.149 ± 0.02 ^c^	0.416 ± 0.03 ^g^	741.92 ± 45.99 ^h^	27.83 ± 1.75 ^a^
*100 mM NaCl + 100 mM KCl + 0.5 mM GB*	Shoot	604.58 ± 13.10 ^fg^	0.881 ± 0.09 ^c^	0.023 ± 0.01 ^b^	0.116 ± 0.02 ^ab^	0.138 ± 0.01 ^bc^	0.339 ± 0.02 ^d^	569.25 ± 41.12 ^g^	97.33 ± 7.97 ^ab^

Means with different letters in the columns are significantly different at *p* < 0.05 level. The roots and shoots have been evaluated independently of each other. Values are averages of four *Triticum* species.

**Table 5 plants-14-00678-t005:** Effect of salt treatments on the enzymatic and non-enzymatic antioxidant defense system and soluble protein in hulled wheat roots.

Salt Treatments	Section	Soluble Protein (µg mL^−1^ Protein)	SOD (U mL^−1^ Protein)	CAT (U mL^−1^ Protein)	GR (U mL^−1^ Protein)	GST (U mL^−1^ Protein)	APX (U mL^−1^ Protein)	PRO (nmol g^−1^ fw)	MDA (nmol g^−1^ fw)
*Control*	Root	316.33 ± 27.04 ^a^	0.972 ± 0.11 ^e^	0.016 ± 0.01 ^ab^	0.089 ± 0.03 ^a^	0.106 ± 0.03 ^ab^	0.279 ± 0.03 ^a^	68.99 ± 11.70 ^a^	19.38 ± 2.16 ^a^
*30 mM NaCl*	Root	372.33 ± 36.96 ^b^	1.047 ± 0.08 ^f^	0.040 ± 0.01 ^d^	0.155 ± 0.05 ^bc^	0.116 ± 0.02 ^b^	0.328 ± 0.02 ^cd^	275.75 ± 68.69 ^cd^	45.67 ± 7.55 ^a^
*30 mM KCl*	Root	442.75 ± 25.24 ^cd^	0.903 ± 0.03 ^cd^	0.031 ± 0.01 ^c^	0.141 ± 0.05 ^bc^	0.108 ± 0.01 ^ab^	0.318 ± 0.02 ^c^	261.58 ± 27.28 ^cd^	40.42 ± 3.82 ^a^
*50 mM NaCl*	Root	392.45 ± 24.45 ^bc^	1.013 ± 0.07 ^ef^	0.046 ± 0.01 ^de^	0.207 ± 0.07 ^d^	0.142 ± 0.02 ^c^	0.361 ± 0.02 ^e^	334.64 ± 60.98 ^d^	62.27 ± 9.87 ^ab^
*50 mM KCl*	Root	491.33 ± 18.40 ^de^	1.012 ± 0.11 ^ef^	0.038 ± 0.02 ^cd^	0.185 ± 0.06 ^cd^	0.127 ± 0.02 ^bc^	0.336 ± 0.03 ^d^	418.00 ± 47.61 ^e^	58.75 ± 8.48 ^ab^
*100 mM NaCl*	Root	391.75 ± 31.26 ^bc^	1.016 ± 0.09 ^ef^	0.040 ± 0.01 ^d^	0.221 ± 0.09 ^d^	0.133 ± 0.02 ^bc^	0.323 ± 0.01 ^cd^	442.25 ± 48.18 ^ef^	179.50 ± 22.73 ^bc^
*100 mM KCl*	Root	469.00 ± 10.73 ^d^	1.028 ± 0.13 ^f^	0.031 ± 0.02 ^c^	0.170 ± 0.06 ^c^	0.120 ± 0.02 ^b^	0.312 ± 0.03 ^c^	511.42 ± 44.04 ^f^	173.50 ± 20.72 ^bc^
*150 mM NaCl*	Root	373.75 ± 40.37 ^b^	0.941 ± 0.06 ^d^	0.032 ± 0.01 ^c^	0.174 ± 0.07 ^c^	0.181 ± 0.09 ^d^	0.299 ± 0.01 ^b^	257.00 ± 17.68 ^c^	283.00 ± 31.27 ^cd^
*150 mM KCl*	Root	450.25 ± 12.14 ^d^	0.965 ± 0.09 ^de^	0.027 ± 0.01 ^bc^	0.148 ± 0.04 ^bc^	0.110 ± 0.02 ^b^	0.306 ± 0.02 ^bc^	372.83 ± 20.86 ^de^	292.00 ± 42.23 ^cd^
*200 mM NaCl*	Root	328.75 ± 21.76 ^a^	0.855 ± 0.05 ^bc^	0.023 ± 0.01 ^b^	0.103 ± 0.03 ^ab^	0.086 ± 0.01 ^a^	0.271 ± 0.01 ^a^	144.92 ± 13.40 ^b^	379.17 ± 23.74 ^de^
*200 mM KCl*	Root	410.17 ± 19.66 ^c^	0.887 ± 0.07 ^c^	0.025 ± 0.01 ^bc^	0.109 ± 0.04 ^ab^	0.097 ± 0.01 ^ab^	0.276 ± 0.03 ^a^	141.08 ± 8.21 ^b^	385.08 ± 26.96 ^de^
*50 mM NaCl + 50 mM KCl*	Root	530.17 ± 31.52 ^ef^	1.179 ± 0.15 ^g^	0.036 ± 0.01 ^cd^	0.210 ± 0.07 ^d^	0.144 ± 0.02 ^c^	0.326 ± 0.01 ^cd^	615.67 ± 32.00 ^gh^	54.58 ± 6.61 ^ab^
*100 mM NaCl + 100 mM KCl*	Root	489.67 ± 14.09 ^de^	0.892 ± 0.07 ^c^	0.027 ± 0.01 ^bc^	0.139 ± 0.06 ^b^	0.094 ± 0.01 ^a^	0.295 ± 0.03 ^b^	592.33 ± 27.29 ^g^	172.17 ± 22.04 ^bc^
*150 mM NaCl + 0.5 mM GB*	Root	534.08 ± 22.44 ^ef^	0.983 ± 0.06 ^e^	0.039 ± 0.01 ^cd^	0.193 ± 0.08 ^cd^	0.115 ± 0.01 ^b^	0.341 ± 0.02 ^d^	411.17 ± 28.00 ^e^	259.00 ± 25.19 ^cd^
*200 mM NaCl + 0.5 mM GB*	Root	506.92 ± 12.43 ^e^	0.904 ± 0.04 ^cd^	0.061 ± 0.11 ^f^	0.123 ± 0.03 ^b^	0.107 ± 0.01 ^ab^	0.317 ± 0.03 ^c^	266.33 ± 26.77 ^c^	317.00 ± 14.35 ^d^
*150 mM KCl + 0.5 mM GB*	Root	546.00 ± 12.27 ^ef^	1.013 ± 0.09 ^ef^	0.034 ± 0.02 ^c^	0.187 ± 0.04 ^cd^	0.141 ± 0.02 ^c^	0.431 ± 0.05 ^h^	463.42 ± 31.60 ^ef^	236.08 ± 21.38 ^c^
*200 mM KCl + 0.5 mM GB*	Root	609.17 ± 14.84 ^fg^	0.964 ± 0.08 ^de^	0.035 ± 0.01 ^c^	0.139 ± 0.05 ^b^	0.121 ± 0.01 ^b^	0.382 ± 0.01 ^ef^	245.92 ± 18.69 ^c^	299.92 ± 28.60 ^d^
*50 mM NaCl + 50 mM KCl + 0.5 mM GB*	Root	667.92 ± 15.38 ^g^	1.252 ± 0.16 ^h^	0.046 ± 0.01 ^de^	0.260 ± 0.07 ^e^	0.179 ± 0.02 ^d^	0.416 ± 0.02 ^g^	748.42 ± 54.23 ^h^	40.58 ± 1.83 ^a^
*100 mM NaCl + 100 mM KCl + 0.5 mM GB*	Root	597.25 ± 10.42 ^fg^	0.897 ± 0.17 ^c^	0.041 ± 0.01 ^d^	0.160 ± 0.07 ^c^	0.128 ± 0.01 ^bc^	0.373 ± 0.01 ^e^	772.67 ± 42.89 ^h^	103.33 ± 7.88 ^b^

Means with different letters in the columns are significantly different at *p* < 0.05 level. Values are averages of four *Triticum* species.

**Table 6 plants-14-00678-t006:** Percentage change of antioxidant enzymes, carotene, and morphological characters under salt treatments compared to control.

Wheats	SOD	CAT	GR	GST	APX	PRO	MDA	Carotene	Fw	Dw	PH	LL
*T. monococcum*	19.3 ^b^	112.90 ^a^	76.20 ^a^	54.45 ^b^	24.80 ^ab^	1099.85 ^b^	1751.75 ^a^	51.85 ^a^	14.74 ^b^	22.71 ^c^	21.35 ^c^	1.52 ^a^
*T. dicoccum*	0.36 ^a^	125.68 ^a^	85.56 ^a^	115.6 ^c^	33.96 ^b^	1079.46 ^b^	1863.06 ^b^	82.68 ^b^	3.19 ^a^	20.51 ^c^	9.10 ^b^	7.95 ^ab^
*T. speltoides*	4.34 ^a^	228.59 ^b^	126.88 ^b^	12.50 ^a^	19.70 ^a^	842.15 ^a^	1830.44 ^ab^	93.77 ^bc^	1.16 ^a^	4.79 ^a^	3.91 ^a^	5.76 ^a^
*T. boeoticum*	14.02 ^b^	231.37 ^b^	195.58 ^c^	0.97 ^a^	29.81 ^ab^	918.78 ^a^	1885.45 ^b^	111.28 ^c^	−3.45 ^a^	12.75 ^b^	1.25 ^a^	11.11 ^b^

Means with different letters in the columns are significantly different at *p* < 0.05 level. Fw: Fresh weight; Dw: Dry weight; PH: Plant height; LL: Leaf length.

**Table 7 plants-14-00678-t007:** Multiple comparisons of hulled wheat species.

Dependent Variable		(I) Wheat	(J) Wheat	Mean Difference (I − J)	Std. Error	*p*	95% Confidence Interval	
							Lower Bound	Upper Bound
Soluble Protein Cons.	LSD	*T. monococcum*	*T. dicoccum*	−5.4912 *	1.24804	0.000	−7.9471	−3.0353
			*T. speltoides*	28.1228 *	1.24804	0.000	25.6669	30.5787
			*T. boeoticum*	26.4211 *	1.24804	0.000	23.9652	28.8769
		*T. dicoccum*	*T. monococcum*	5.4912 *	1.24804	0.000	3.0353	7.9471
			*T. speltoides*	33.6140 *	1.24804	0.000	31.1582	36.0699
			*T. boeoticum*	31.9123 *	1.24804	0.000	29.4564	34.3682
		*T. speltoides*	*T. monococcum*	−28.1228 *	1.24804	0.000	−30.5787	−25.6669
			*T. dicoccum*	−33.6140 *	1.24804	0.000	−36.0699	−31.1582
			*T. boeoticum*	−1.7018	1.24804	0.174	−4.1576	0.7541
		*T. boeoticum*	*T. monococcum*	−26.4211 *	1.24804	0.000	−28.8769	−23.9652
			*T. dicoccum*	−31.9123 *	1.24804	0.000	−34.3682	−29.4564
			*T. speltoides*	1.7018	1.24804	0.174	−0.7541	4.1576
SOD	LSD	*T. monococcum*	*T. dicoccum*	0.0252 *	0.00798	0.002	0.0095	0.0410
			*T. speltoides*	−0.1425 *	0.00798	0.000	−0.1582	−0.1268
			*T. boeoticum*	−0.0981 *	0.00798	0.000	−0.1138	−0.0824
		*T. dicoccum*	*T. monococcum*	−0.0252 *	0.00798	0.002	−0.0410	−0.0095
			*T. speltoides*	−0.1677 *	0.00798	0.000	−0.1835	−0.1520
			*T. boeoticum*	−0.1233 *	0.00798	0.000	−0.1390	−0.1076
		*T. speltoides*	*T. monococcum*	0.1425 *	0.00798	0.000	0.1268	0.1582
			*T. dicoccum*	0.1677 *	0.00798	0.000	0.1520	0.1835
			*T. boeoticum*	0.0444 *	0.00798	0.000	0.0287	0.0601
		*T. boeoticum*	*T. monococcum*	0.0981 *	0.00798	0.000	0.0824	0.1138
			*T. dicoccum*	0.1233 *	0.00798	0.000	0.1076	0.1390
			*T. speltoides*	−0.0444 *	0.00798	0.000	−0.0601	−0.0287
CAT	LSD	*T. monococcum*	*T. dicoccum*	0.0001	0.00288	0.971	−0.0056	0.0058
			*T. speltoides*	−0.0169 *	0.00288	0.000	−0.0225	−0.0112
			*T. boeoticum*	−0.0165 *	0.00288	0.000	−0.0221	−0.0108
		*T. dicoccum*	*T. monococcum*	−0.0001	0.00288	0.971	−0.0058	0.0056
			*T. speltoides*	−0.0170 *	0.00288	0.000	−0.0226	−0.0113
			*T. boeoticum*	−0.0166 *	0.00288	0.000	−0.0222	−0.0109
		*T. speltoides*	*T. monococcum*	0.0169 *	0.00288	0.000	0.0112	0.0225
			*T. dicoccum*	0.0170 *	0.00288	0.000	0.0113	0.0226
			*T. boeoticum*	0.0004	0.00288	0.886	−0.0053	0.0061
		*T. boeoticum*	*T. monococcum*	0.0165 *	0.00288	0.000	0.0108	0.0221
			*T. dicoccum*	0.0166 *	0.00288	0.000	0.0109	0.0222
			*T. speltoides*	−0.0004	0.00288	0.886	−0.0061	0.0053
GR	LSD	*T. monococcum*	*T. dicoccum*	0.0014	0.00336	0.679	−0.0052	0.0080
			*T. speltoides*	−0.0877 *	0.00336	0.000	−0.0943	−0.0811
			*T. boeoticum*	−0.0715 *	0.00336	0.000	−0.0781	−0.0649
		*T. dicoccum*	*T. monococcum*	−0.0014	0.00336	0.679	−0.0080	0.0052
			*T. speltoides*	−0.0891 *	0.00336	0.000	−0.0957	−0.0825
			*T. boeoticum*	−0.0729 *	0.00336	0.000	−0.0795	−0.0663
		*T. speltoides*	*T. monococcum*	0.0877 *	0.00336	0.000	0.0811	0.0943
			*T. dicoccum*	0.0891 *	0.00336	0.000	0.0825	0.0957
			*T. boeoticum*	0.0162 *	0.00336	0.000	0.0096	0.0228
		*T. boeoticum*	*T. monococcum*	0.0715 *	0.00336	0.000	0.0649	0.0781
			*T. dicoccum*	0.0729 *	0.00336	0.000	0.0663	0.0795
			*T. speltoides*	−0.0162 *	0.00336	0.000	−0.0228	−0.0096
GST	LSD	*T. monococcum*	*T. dicoccum*	−0.0189 *	0.00623	0.003	−0.0312	−0.0067
			*T. speltoides*	−0.0432 *	0.00623	0.000	−0.0554	−0.0309
			*T. boeoticum*	−0.0242 *	0.00623	0.000	−0.0364	−0.0119
		*T. dicoccum*	*T. monococcum*	0.0189 *	0.00623	0.003	0.0067	0.0312
			*T. speltoides*	−0.0242 *	0.00623	0.000	−0.0365	−0.0120
			*T. boeoticum*	−0.0052	0.00623	0.404	−0.0175	0.0071
		*T. speltoides*	*T. monococcum*	0.0432 *	0.00623	0.000	0.0309	0.0554
			*T. dicoccum*	0.0242 *	0.00623	0.000	0.0120	0.0365
			*T. boeoticum*	0.0190 *	0.00623	0.002	0.0067	0.0313
		*T. boeoticum*	*T. monococcum*	0.0242 *	0.00623	0.000	0.0119	0.0364
			*T. dicoccum*	0.0052	0.00623	0.404	−0.0071	0.0175
			*T. speltoides*	−0.0190 *	0.00623	0.002	−0.0313	−0.0067
APX	LSD	*T. monococcum*	*T. dicoccum*	0.0207 *	0.00121	0.000	0.0183	0.0231
			*T. speltoides*	−0.0272 *	0.00121	0.000	−0.0296	−0.0248
			*T. boeoticum*	−0.0127 *	0.00121	0.000	−0.0151	−0.0104
		*T. dicoccum*	*T. monococcum*	−0.0207 *	0.00121	0.000	−0.0231	−0.0183
			*T. speltoides*	−0.0479 *	0.00121	0.000	−0.0503	−0.0455
			*T. boeoticum*	−0.0334 *	0.00121	0.000	−0.0358	−0.0311
		*T. speltoides*	*T. monococcum*	0.0272 *	0.00121	0.000	0.0248	0.0296
			*T. dicoccum*	0.0479 *	0.00121	0.000	0.0455	0.0503
			*T. boeoticum*	0.0145 *	0.00121	0.000	0.0121	0.0169
		*T. boeoticum*	*T. monococcum*	0.0127 *	0.00121	0.000	0.0104	0.0151
			*T. dicoccum*	0.0334 *	0.00121	0.000	0.0311	0.0358
			*T. speltoides*	−0.0145 *	0.00121	0.000	−0.0169	−0.0121
Proline	LSD	*T. monococcum*	*T. dicoccum*	−30.1632 *	1.32946	0.000	−32.7793	−27.5470
			*T. speltoides*	−68.0053 *	1.32946	0.000	−70.6214	−65.3892
			*T. boeoticum*	−52.9614 *	1.32946	0.000	−55.5775	−50.3453
		*T. dicoccum*	*T. monococcum*	30.1632 *	1.32946	0.000	27.5470	32.7793
			*T. speltoides*	−37.8421 *	1.32946	0.000	−40.4582	−35.2260
			*T. boeoticum*	−22.7982 *	1.32946	0.000	−25.4144	−20.1821
		*T. speltoides*	*T. monococcum*	68.0053 *	1.32946	0.000	65.3892	70.6214
			*T. dicoccum*	37.8421 *	1.32946	0.000	35.2260	40.4582
			*T. boeoticum*	15.0439 *	1.32946	0.000	12.4277	17.6600
		*T. boeoticum*	*T. monococcum*	52.9614 *	1.32946	0.000	50.3453	55.5775
			*T. dicoccum*	22.7982 *	1.32946	0.000	20.1821	25.4144
			*T. speltoides*	−15.0439 *	1.32946	0.000	−17.6600	−12.4277
MDA	LSD	*T. monococcum*	*T. dicoccum*	8.3553 *	1.27725	0.000	5.8419	10.8686
			*T. speltoides*	−28.1500 *	1.27725	0.000	−30.6634	−25.6366
			*T. boeoticum*	−19.8781 *	1.27725	0.000	−22.3914	−17.3647
		*T. dicoccum*	*T. monococcum*	−8.3553 *	1.27725	0.000	−10.8686	−5.8419
			*T. speltoides*	−36.5053 *	1.27725	0.000	−39.0186	−33.9919
			*T. boeoticum*	−28.2333 *	1.27725	0.000	−30.7467	−25.7200
		*T. speltoides*	*T. monococcum*	28.1500 *	1.27725	0.000	25.6366	30.6634
			*T. dicoccum*	36.5053 *	1.27725	0.000	33.9919	39.0186
			*T. boeoticum*	8.2719 *	1.27725	0.000	5.7586	10.7853
		*T. boeoticum*	*T. monococcum*	19.8781 *	1.27725	0.000	17.3647	22.3914
			*T. dicoccum*	28.2333 *	1.27725	0.000	25.7200	30.7467
			*T. speltoides*	−8.2719 *	1.27725	0.000	−10.7853	−5.7586

Based on observed means. The error term is Mean Square (Error) = 92.988. The mean difference is significant at the 0.05 level.

## Data Availability

All data generated or analyzed during this study are included in this published article (and its Appendix A).

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
