# Peer review of "Soil Salinization and Ancient Hulled Wheat: A Study on Antioxidant Defense Mechanisms"

_plants, 2025, doi:10.3390/plants14050678_

Round 1
Reviewer 1 Report (Previous Reviewer 1)
Comments and Suggestions for Authors
Dear Editor,
The manuscript entitled Soil Salinization and Ancient Hulled Wheat: A Study on Antioxidant Defense Mechanisms presents an interesting data set on the effects of individual or combinated salt treatment use on salinity mitigation in a species of food interest. The topic is appropriate for the journal and the text is well written. However, some changes should be first made in the original version, to improve several technical and formal aspects of the manuscript, according to the following suggestions:
I reread the changes made to the manuscript and noticed a new paragraph with the effect of salt stress on plants with reference to wheat species from Turkey which is an added value for the manuscript.
Pay attention, in table 3 you put total protein but in fact it is soluble protein according to the material and method section
As I mentioned in the other review of this work, I do not understand to which wheat species (out of the 4 studied) the values in tables 1 and 2 belong. The same question for tables 3 and 4. What does it mean - Values are averages of four Triticum species - did you measure the dimensions of the 4 species and average them?
Pay attention, in figure 5 you must leave a free space for the following information
My conclusion is that these results are very valuable, interesting also can also open new perspectives in understanding salt-stress mechanisms, as well as possible ways to alleviate their negative effects. In this way, this kind of research may be also promising for economical point of view, in the present context of salinization in agriculture and the need to find more salt-tolerant plants to support a continuously growing population.
Therefore, I think that the manuscript is suitable for publication in the present form after some low modifications.
Author Response
Responses for referee 1
- In Table 3, the expression “total protein” was changed to “soluble protein”.
- The expressions given in Table 1-4 are given by taking the average of the values obtained from 4 hulled wheat varieties. Individual data for hulled wheats are also given in the supplementary materials.
- The statements given in the explanations of Figure 5 are explanations of the figure. For this reason, that expression was put in parentheses to eliminate confusion.

Reviewer 2 Report (Previous Reviewer 4)
Comments and Suggestions for Authors
Given that the author has considered (most of) the received comments/suggestions, the new version of the manuscript is significantly improved. To be published, some minor revisions are needed.
-Add references to the text from lines 75-88.
-Given the very large size (36 pages) of the manuscript, and the requirement that a Figure must be placed near the related text, I consider that you must transfer the Figures 1-2 to supplementary materials. It will be very difficult for the readers to understand your results (for instance Effects of Salt Stress on Antioxidant Enzyme Activity in Plant Roots and Stems) when the associated comments are on page 8 and the figure with the results is on page 4.
-In lines 180-182 there are no comments regarding the results for the effects of salt stress on antioxidant enzyme activity in plant roots and stems, as such I consider that you can remove this text.
-Transfer Table 7 from Discussion to Results section.
-Separate the text from lines 526-544 from the title of Figure 5. In this format, the text is considered part of the title.
Author Response
Responses for referee 2
- The statements between lines 75-78 are a text that was previously in the conclusion section and was included in the introduction section as a result of the referee evaluation. Since this text consists entirely of my own statements, no references have been added.
- Taking into consideration the evaluations of other referees, figures 1 and 2 were not included in the additional materials in order not to impair the understandability of the study.
- Lines 180-182 were removed from the manuscript.
- Table 7 was moved from the Discussion section to the Results section.
- The statements given in the explanations of Figure 5 are explanations of the figure. For this reason, that expression was put in parentheses to eliminate confusion.

Reviewer 3 Report (New Reviewer)
Comments and Suggestions for Authors
Salt stress is a major factor, which restricts plant growth and reduces crop yields. The tolerances among varieties are different. This manuscript evaluated the salt tolerance of four ancient hulled wheat species.
Major questions.
- The research was not well designed. Different factors were selected for the hydroponic experiment. For KCl, NaCl, and glycine-betaine treatment, which is the real control? It is difficult to understand.
- There are so many data in Table 1 and Table 2. But Table 1 and 2 are not well designed. Shoot or stem and root should be separated exhibited.
- The results are not well written. The results could not answer the research aims.
Minor questions.
See attachment.

The English presentation does not affect comprehension
Author Response
Responses for referee 3
- Three culture pots from each of the four studied hulled wheat varieties were grown as controls without any stress or glycine-betaine application. These were fed only with Hoagland solution. Therefore, the control of each wheat variety consists of plants grown without any stress or GB application. The data obtained from these control plants are given in Tables 1-4 by taking the averages (data is given by taking the averages in the same way in other stress and GB applications).
- Tables 1-4 have been reorganized to separate root and shoot data.
- The paragraph order in the Introduction section has been rearranged.
- We chose the term "stem" to generally represent the above-ground body, which includes the leaves. The term "Shoot" could also be preferred over the term "stem", but we preferred the term "stem" because we thought it would be more appropriate. In general, both terms mean the same thing.
- To evaluate the impact of salt stress on different wheat varieties and plant parts, a comprehensive statistical analysis was conducted using two-way (salt doses and wheat varieties, as well as salt doses and plant parts) and three-way analysis of variance (ANOVA) (examined the combined effects of salt doses, plant parts (root and leafy stem), and wheat varieties ( monococcum, T. dicoccum, T. speltoides, and T. boeoticum)).
- The images in Figure 1 a, b, and d are given to show the averages of the data obtained from 4 wheat varieties.
- Plant Height: It gives the total length of the plant above the ground. Leaf Height: It gives the length of the longest leaf emerging from the stem.
- In Figure 4, when all the text given in the figure descriptions are shown on the figure, the figure becomes very complicated. Therefore, in order to preserve the simplicity of the figure, the texts are given in abbreviations and the explanations are given under the figure. However, some adjustments have been made to Figure 4.
- The heading 3.1 has been changed to "Effects of Salt Stress on Plant Growth".
- My laboratory does not have the necessary equipment to determine the K+/Na+ ratio. Since I do not have sufficient financial resources to have this analysis done for a fee, this analysis could not be performed.
- Figure 5 includes an interpretation of the data obtained in this study based on previous studies. The figure was drawn by me personally, and the sources used in drawing the figure and the necessary explanations are given with their references in the explanation section below the figure. The statements given in the explanations of Figure 5 are explanations of the figure. For this reason, that expression was put in parentheses to eliminate confusion.
- The phrase "20 seeds in each culture pot" has been added to subheading 4.1. Twenty seeds were placed in each culture container. Culture vessels were prepared three for each application. For example, for 1 set of experiments, a total of 228 (19 applications * 3 culture vessels * 4 wheat varieties) culture dishes were prepared, three for control, three for 30 mM NaCl application, three for 50 mM NaCl application, etc. These experimental conditions were repeated three times independently. Each set of experiments lasted 25 days with 10 days of stress-free cultivation followed by 15 days of stress treatment. Trials were performed independently for a total of three sets (25 * 3 = 75 days). As a result, nine culture vessels were obtained from each application. Plants from three culture vessels grown in each set of experiments were harvested and combined, and the average of the three culture vessels was recorded as one replicate (There are data for three culture vessels in one replicate, twenty wheat in each vessel, sixty wheat sprouts in total).

Round 2
Reviewer 3 Report (New Reviewer)
Comments and Suggestions for Authors< !--StartFragment -->
Shoot and stem are different. A shoot is a visible part of a plant that bears flower buds, lateral buds, and flowering stems.
A stem is a plant axis that is made by some nodes and so many internodes.
< !--EndFragment -->
The Statistics related content has not revised sufficiently in tables and result section.
< !--StartFragment -->
What is this manuscript’s founding? This should be explained in the figure 5.
< !--EndFragment -->
Author Response
- The necessary explanation for Figure 5 has been added after the figure description.
(In the schematic representation of the antioxidative defense response that develops under salt stress in hulled wheat, data presented in the studies of Yang and Guo [1], Zhu [51], Jiang et al [65], Lin et al [66], Chung et al [67], Verslues et al [68], and Jabeen et al [69] were used).
However, the statement given in the explanation section of Figure 5 has been included in the text.
(Sodium ions are initially sensed by sensors localized in the plasma membrane (PM). Salt stress induces ionic stress, which results in changes in the calcium status of the cytosol [1]. Glycosyl inositol phosphorylceramides (GIPCs) are abundant in the PM and receive these signals [65]. SOS2 kinase is induced by sodium [66], while salt stress induces ROS stress [1], which in turn modulates the transcription level of SOS1 [67]. By interacting with CAT, SOS2 connects the SOS pathway to other signaling pathways and phosphorylates and activates SOS1 [68]. On the other hand, GIPC increases calcium signaling by binding to Na+, while calcium receptors bind to intracellular Ca2+ and activate Na+/H+ antiporter activity. MPK6, activated by phosphatidic acid (PA), phosphorylates SOS1 and increases its activity. FER senses changes in the cell wall under salt stress and mediates calcium signaling for long-term stress. ANNEXINs (ANNs) modulate calcium signaling under salt stress, promoting activation of SOS2 activity by SCaBP8. ROS released under stress activate the enzymatic and non-enzymatic antioxidant defense system. Many osmoregulators, carotenes, alkaloids, flavonoids, tocopherols, phenolic compounds, non-protein amino acids and a number of yet unidentified metabolites are activated to support antioxidant defense. The SOS pathway and the released ROS force gene regulatory systems to come into play. Following all these, the ROS released in the organism are neutralized, ionic homeostasis and subsequently cellular stress resistance are achieved [51]).
- Further explanations are provided in the text regarding the statistical data given in the tables. The added texts are given below.
(Levene’s Test is a statistical test used to evaluate the homogeneity of data. Tests the null hypothesis that the error variance of the dependent variable is equal across groups. The result of the Levene test allows us to determine whether the difference between groups is statistically significant. If the difference is significant as a result of the Levene test, it means that there is an inequality of variance between the groups and in this case, we may need to use different analysis methods. In our study, the p values of all tested variables were greater than 0.05, which suggests that there is no variance inequality between the groups and that no other testing tool is needed.) MANOVA Multivariate test creates four test statistics (Pillai's Trace, Wilks' Lambda, Hotelling's Trace, Roy's Largest Root) and if the p ≥ 0.05 in the tested factors, these test statistics are looked at. In this study, although the F values for each test statistic change, the null hypothesis of MANOVA is rejected because the p ≤ 0.05 (p ≤ 0.001) and it is concluded that both wheat varieties, plant parts and salt stress doses are important in the enzymatic/non-enzymatic antioxidative response (Table S2.3).
- The word “stem” in the manuscript has been replaced with the word “shoot”.
- Table 6 has been moved two paragraphs up in the text.

Round 3
Reviewer 3 Report (New Reviewer)
Comments and Suggestions for Authors
Too much references. The questions in report 2 were not revised or answered.
This manuscript is a resubmission of an earlier submission. The following is a list of the peer review reports and author responses from that submission.
Round 1
Reviewer 1 Report
Comments and Suggestions for Authors
Specific comment
Abstract: concise, descriptive, emphasizing the idea, aim and results of the present study.
Observations:
Perhaps it would be good to modified in line 16 - T. boeoticum with Triticum boeoticum
In addition, perhaps it would be good for the author to indicate which wheat species he worked with because it would be clear that only T. boeoticum was used.
Introduction: very well written, well documented with relevant literature and it is able to introduce and familiarize with the subject of the study, in a logical way.
Observations:
Perhaps it would be good to put the phrase Salt stress, a major abiotic stressor, induces oxidative stress by increasing ROS production [19, 20] before the one in line 58 so that it is not an abrupt transition from one paragraph to another.
Results: are presented in a concise, easy to follow manner, also using the tables and figures, which are well organized.
Observations:
It is not clear in table 1, 2,3,4 which species you worked with.
There is a lot of data but the tables are presented chaotically which does not allow a quick follow-up.
Subchapter 2.2. - you determined soluble proteins by the Bradford method and not total protein. So modify it with soluble protein
In table 3 and 4 total protein is not an antioxidant and has no place there or you can modify the title by adding this word
In fact, it should be changed throughout the text to proline content not activity.
Pay attention at line 483 and please mention the species the authors worked with.
Discussions enclosed precise comments based on obtained results. Throughout this section, many references are being used, in order to support the given comments.
Observations:
Pay attention in figure 5, specify in the title which graphs are for the root and which are for the stem
At line 453 pay attention and modified proline content and put activities after CAT
Materials and methods: clear, logical and accurate.
Observations:
Pay attention at section 4.3. line 584 and modified concentration with level. In addition, the results were expressed as mg /g fresh weight or μg?
In the formula on line 606, and 632 what does prot cons represent?
Tables S1.1. - S1.4. should be included in the text and not in the annexes.
The conclusions formulated by the author are general and do not indicate the results obtained in this extensive presentation of the data. In addition, he uses data from the literature which is not indicated in the bibliography.

Author Response
- In line 16, boeoticum has been changed to Triticum boeoticum.
- Since the wheat varieties used were clearly stated in the material and method section, there was no need to rewrite them here.
- The sentence "Salt stress, a major abiotic stressor, induces oxidative stress by increasing ROS production [19, 20]" is moved before line 58.
- The statement "Data are given as the average of the four wheat varieties studied" is added below each table.
- The placement of the tables in the manuscript has been rearranged.
- The term “total protein” has been replaced with soluble protein.
- The phrase "total protein" has been added to the table title in Table 3 and 4.
- This term has been changed to "proline accumulation" throughout the manuscript.
- In line 483, the variety name that the researchers studied is added.
- Explanatory information has been added under Figure 5.
- Enzyme activity increases are included in the text. (276 mg gr-1 DW, 7.05 mg gr-1 DW, 7.60 mg gr-1 DW and 65.69 mg gr-1 DW, respectively).
- In section 4.3, the incorrectly written expression as µg ml-1 protein has been corrected to mg ml-1 protein.
- The expression in line 606 has been changed to "units per milligram of protein" to ensure standardization.
- Since there are too many tables and figures in the text, I do not find it appropriate to place the tables in the supplementary materials suggested by the referee (S1.1-S1.4, 4 tables) in the text in order not to spoil the visuality.
- The text description that should have been in the description section of Figure 4 was accidentally forgotten. The necessary explanation was added to that section, and the necessary references were added.

Reviewer 2 Report
Comments and Suggestions for Authors
In my opinion, the scope of the study does not bring the required novel aspects to qualify the article for further evaluation stages. The parameters determined are very basic. Given the current advanced state of knowledge in the field of plant biology, the determination of these basic parameters in the study is not sufficient to be able to recommend a revision of the manuscript. In my opinion, the manuscript does not meet the substantive requirements. In addition, it lacks a clearly defined research problem.
Author Response
The spelt wheats I used in my article were used intensively in Turkey, the homeland of wheat, almost a century ago, but have fallen behind in competition with today's modern wheats due to various preferential reasons and are cultivated in limited areas. Reintroducing forgotten genetic resources to agriculture is also an innovation. Combating global climate change, one of today's biggest problems, can be done not only by developing new methods, but also by bringing our forgotten or nearly forgotten genetic resources back into use. For this reason, I do not agree with the view that my article does not contain any innovation.
However, I also disagree with the idea that my article does not contain a research problem. My article has a research problem: What are the responses of old spelt wheat to adverse conditions caused by climate change? Antioxidant enzyme activities cover the first responses of plants to adverse conditions. Adverse climatic conditions create stress in plants, and this resulting stress triggers ROS production in plants. These released ROS act destructively and disruptively in plants. The ability of the plant to cope with these cellular irregularities can again be measured by its antioxidant capacity. Making a preliminary decision by looking at antioxidant enzyme capacities offers us great advantages when compared to long field trials and high financial costs in agricultural production. Therefore, short-term laboratory trials before field trials provide advantages in terms of selectivity in breeding and time and cost savings.

Reviewer 3 Report
Comments and Suggestions for Authors
Dear author,
the paper could not be published in Plants because you previously published the same results as a preliminary study (Antioxidant defense responses of hulled wheat varieties to the addition of sodium and potassium salts and exogenous glycinebetaine, and evaluation of the usability of these hulled wheats in the remediation of saline soils).
Author Response
Yes, I have previously published my article as a preprint and I have conveyed this to you in my cover letter when submitting my article to your journal. Your journal also encourages publishing as a preprint. Therefore, I believe this does not prevent my article from being evaluated. I am giving the preprint address of my article again below.
https://doi.org/10.21203/rs.3.rs-4368507/v1

Reviewer 4 Report
Comments and Suggestions for Authors
In this study the author evaluates the enzymatic and non-enzymatic antioxidant defense mechanisms of four ancient hulled wheat species under salt stress, with and without exogenous application of glycine-betaine. The research represents an extensive study that involved a large volume of measurements and analyses. However, I regret to recommend that the manuscript is not acceptable for publication in its present form. All detailed comments and suggestions can be found bellow:
- I recommend that you arrange the keywords alphabetically.
- In the abstract (lines 10, 11, 16) as well in the text, I consider necessary to replace “ancient hulled wheat varieties” with ancient hulled wheats or ancient hulled wheat species, given that those so-called four (T. monococcum L., T. boeoticum Boiss, T. dicoccum Schrank, and T. speltoides) varieties are actually species. The term variety is used in botany for a taxonomic unit that ranks below subspecies or species. As well it is used for cultivated varieties (cultivar).
- The text from lines 51-57 has no reference(s).
-For the values inside the Tables, replace comma (,) with dot (.)
- Replace “Fig.” with Figure.
- In the titles of Tables 1-2 add… in hulled wheats. Also, explain in the footer of Tables 1-4 values are average of four Triticum species.
- The expression “*Differences in the letters indicates statistical significance at 5% level in the columns.” is duplicated in lines 118-119. I recommend you to replace this expression with Means with different letters in the columns are significantly different at p<0.05. Also in line 118 replace FW with Fw, and DW with Dw; remove “GB: Glycine-Betaine”, because this abbreviation was not used in Table 1. Insert “GB: Glycine-Betaine”, in the footer of Table 2.
- In the title of Tables 1-5, and Tables S1.1-7, replace “salt applications” with salt treatments. Also, replace “salt applications” with salt treatments in the first column of Tables 1-4 and Tables S1.1-7. Replace in the footer of Tables S1.1-7 “*Differences in the letters indicates statistical significance at 5% level in the columns.” with Different letters in the columns indicates significant differences (at p<0.05) between wheat species.
-Explain in the M&M the origin/provenance of the biological material. Given that you stated in line 75 “ancient Turkish hulled wheat”, are these landraces still cultivated in some region of Turkey?
- In the M&M subsection 3.1, give information regarding the size/volume of growing containers and composition of the used substrate. Also, explain how the salt treatments were applied: daily/at different period; dose/quantity of salt solution applied per plant?
- In order to increase the citation potential of your study, I consider that is necessary to express the salt treatment solutions in electrical conductivity-EC (dS/m). Also, you must indicate the EC of used Hoagland solution.
-Why did you evaluate the effect of salt stress after 15 days and not after a longer period of stress?
- In Tables 1-2, and Table S1.2-3, replace “leaf height” with leaf length.
- In Tables 3-5 replaces “ml” with mL.
-In Tables S1.4-7, insert the measurement units for different compounds.
- Reformulate the title of subsection 2.6, as follows: ANOVA, MANOVA and Levene's test results.
- In the Discussion section do not mention numbers of Tables and Figures.
- Given the (very) large size of the manuscript, and that the Discussion section (12.5 pages) is larger that Results section (9.5 pages), you must transfer Figures 1-4, 6 and Table 5 (along with the comments associated with their data) from Discussion to Results section.
- In Figure 1, each graph must have a different title, and not identical with the Figure title. Do the same for graphs from Figure 2. Explain the abbreviation Tm, Ts in the footer of Figure 1.
- The title of Table S2.1 is multiple comparisons, but you made only comparisons of einkorn with other three Triticum species, why?
-In the header of Table S2.1-4, replace Sig (Significance-probably) with p-value.
-Lines 529-530, reformulate the text “Statistical analysis revealed no significant differences between varieties in terms of CAT and GR enzyme activities” as follows: Statistical analysis revealed no significant differences between einkorn (T.monococcum) and other wheat species in terms of CAT and GR enzyme activities; given that in this Table only the comparisons of einkorn with other species are presented (not all possible comparisons between these four species).
- Remove Figure 5, given that this data regarding proline and MDA are included in Tables S1.4-7.
- What does N (12/114) represent in Table 5, number of observations/measurements?
- In the footer of Table 5 explain the associated abbreviation: FW Fresh weight; DW Dry weight; PH Plant height; LL Leaf length.
- Total length (TL) from Table 6 is the same with plant height (PH) from Table 5. Check?
- At the end of title for Figure 4 insert its associated reference [48].
- Do not begin the title of a table or figure with a symbol (%). Reformulate these titles: Percentage change of traits/compounds under salt stress compared to control.
- Transfer the entire text of subsection 3.6 to Results section. Also I recommend you to transfer the Figure 6 supplementary materials, given that the comments associated to these 8 graphs are limited (only 6.5 rows).
Explain the abbreviation T1-18, and 1-8. You must use a title for each biplot. Change the direction of y axis name (PC2), from bottom to top.
- The Conclusions section must be radically restructured. You must formulate only conclusions supported by your results. Include some further/perspective direction of this research.
- Transfer the text “Salt stress poses a significant threat to global wheat production and food security. To address this challenge, developing salt-tolerant wheat varieties and implementing appropriate agronomic practices are crucial. Ancestral hulled wheat varieties, with their inherent salt tolerance, offer a promising avenue for breeding programs. Türkiye, as a wheat gene center, possesses a rich genetic diversity that may harbor salt-tolerant varieties. These varieties, when cultivated in sodic/saline soils, could potentially outperform modern wheat varieties in terms of yield. Leveraging these genetic resources is essential for ensuring food security in increasingly saline agricultural lands.’ from line 700-706 to Introduction section.
- Transfer the text “Wheat, a staple food globally, is vulnerable to salinity stress. Hulled wheat, the ancient ancestor of modern wheat, offers a valuable genetic resource. Modern wheat, bred for specific traits like yield and nutritional quality, may have lost some of its natural salt tolerance.” from lines 713-713. to Introduction or Discussion section.
- Do the same for the text “Moreover, there is growing consumer interest in ancient grains, including hulled wheat, due to their potential health benefits. These grains offer a healthier nutritional profile and may be better tolerated by individuals with gluten sensitivities.” from lines 718-721 to Introduction section.
- Transfer the text from lines 723-729 to Discussion section.
Author Response
- Keywords are listed alphabetically.
- The phrase "ancient hulled wheats" has been changed to "ancient hulled wheat species".
- "United Nations Department of Economic and Social Affairs, Population Division (2021). Global Population Growth and Sustainable Development. UN DESA/POP/2021/TR/NO. 2. Pp. 103-108" has been added as a reference (reference 97) for the text between 51-57.
- The ","s in the tables have been replaced with ".".
- "Fig." has been changed to "Figure".
- The phrase "in hulled wheat" has been added to Tables 1 and 2.
- The following statement has been added at the bottom of Tables 1-4: "Values are averages of four Triticum species."
- The repeated phrase "Differences in the letters indicates statistical significance at 5% level in the columns" has been corrected.
- The expression "Differences in the letters indicates statistical significance at 5% level in the columns" has been changed to "Means with different letters in the columns are significantly different at p<0.05 level".
- In line 118, FW is changed to Fw and DW is changed to Dw. GB description was removed from Table 1 and added to Table 2.
- The phrase "Salt applications" in the tables and table headings has been replaced with "salt treatments". The statement "Different letters in the columns indicate significant differences among wheat types (p<0.05)" has been added to the footer of Tables S1.1-7.
- In the material-method section, information about the origins of the wheat used was added. Yes, these wheats are still grown in limited quantities in certain regions of Diyarbakır, Tunceli, Bitlis, Kayseri and Kastamonu, Türkiye.
- Necessary information has been added in Section 4.1.
- The EC value of the Hoagland solution has been added (pH 6.8; EC: 3.1 dS m-1). However, since I do not have the salt solutions used in the study, their EC values have not been added.
- When salt concentrations of 150 and 200 mM are applied for more than 15 days, they cause serious stress in plants and the plants die. This is based on data obtained in our previous studies. Therefore, a 15-day time limit was set in order to correctly obtain the enzymatic responses of the plant under high salt stress.
- In Table 1-2 and Table S1.2-3, the term "leaf height" has been replaced with "leaf length".
- In Tables 3-5, "mL" is used instead of "ml."
- Added units of measurement for different compounds to Tables S1.4-7.
- The title of subheading 2.6 has been changed to ANOVA, MANOVA and Levene test results
- Removed notifications regarding figures and tables from the discussion section.
- Figures 1-3, along with data from 6 and Table 5, were moved from the discussion section to the results section. However, figure 4 was left in this section because it was relevant to the discussion section (figure number 4 was replaced by 5).
- The content titles of Figures 1 and 2 were changed, and the names of the abbreviations were added to the description section under the figure.
- Multiple comparison tests were performed for all varieties, but since there are dozens of pages of data, only the data for T. monococcum is included here. Multiple comparison data for other varieties can also be shared upon request.
- In the header of Table S2.1-4, p-value was replaced with Sig (Significance-probably).
- The sentence in lines 529-530 has been changed to "Statistical analysis revealed no significant differences between einkorn ( monococcum) and other wheat species in terms of CAT and GR enzyme activities".
- Figure 5 has been removed.
- In Table 5, N12; 4 wheat species ( monococcum, T. dicoccum, T. boeoticum, T. speltoides) * 3 replications = 12
- N114; 2 sections (root and stem) * 19 doses (Control, 30 mM NaCl, 50 mM NaCl, 100 mM NaCl, 150 mM NaCl, 200 mM NaCl, 30 mM KCl, 50 mM KCl, 100 mM KCl, 150 mM KCl, 200 mM KCl, 50 mM NaCl + 50 mM KCl, 100 mM NaCl + 100 mM KCl, 150 mM NaCl + 0.5 mM GB, 200 mM NaCl + 0.5 mM GB, 150 mM KCl + 0.5 mM GB, 200 mM KCl + 0.5 mM GB, 50 mM NaCl + 50 mM KCl + 0.5 mM GB, 100 mM NaCl + 100 mM KCl + 0.5 mM GB) * 3 replications = 114
- Abbreviations added to the footer of Table 5.
- Total height (TL) in Table 6 and plant height (PH) in Table 5 were arranged to be the same (PH).
- Added relevant references (1, 48, 93-96) at the end of Figure 4 (Figure 5 in the new layout).
- Table and figure captions have been rearranged.
- Subsection 3.6 was transferred to the results section with its content. Table S1.1 from supplementary materials was also added to this section (S1.1 was replaced with a multiple comparison table with all wheat species).
- Necessary edits were made to Figure 4 (Figure 6 before the edit) and figure abbreviations.
- Necessary edits and additions have been made in the conclusion section.
- The changes made are shown in yellow on the revised manuscript.
